ecology/environmental science/limnology

velocity, turbulence, aquaculture, recirculating flow, air micro-bubbles

**Author for correspondence:**
Xiangju Cheng
e-mail: chengxiangju@scut.edu.cn

# Hydrodynamics of an in-pond raceway system with an aeration plug-flow device for application in aquaculture: an experimental study

Wuhua Li[1,2], Xiangju Cheng[1,2], Jun Xie[3], Zhaoli Wang[1] and Deguang Yu[3]

[1]State Key Laboratory of Subtropical Building Science, South China University of Technology, Guangzhou 510640, People's Republic of China
[2]State Key Laboratory of Hydraulics and Mountain River Engineering, Sichuan University, Chengdu 610065, People's Republic of China
[3]Pearl River Fisheries Research Institute, Chinese Academy of Fishery Science, Guangzhou 510380, People's Republic of China

XC, 0000-0003-4276-5856

An in-pond raceway system (IPRS) is an effective intensive aquaculture practice for regions with high water consumption and limited land resources. Water flow and dissolved oxygen (DO) are important for sustainable aquaculture. Several innovations have been made in IPRS design and operation to increase water exchange and DO concentration; one of these is the aeration plug-flow device (APFD). The APFD is commonly used in China as the only power source for water recirculation in aquaculture ponds. Understanding of the hydrodynamics of the system is necessary to improve the design of the IPRS with APFD. To this end, we performed experimental studies on a model system. We measured three-dimensional velocity at various locations using an Acoustic Doppler Velocimeter. Velocity distribution and turbulence characteristics were assessed, and plug-flow characteristics were analysed. Two patterns of velocity and turbulence in horizontal sections were observed: near the APFD, the water flow was intensively pushed downstream and simultaneously recirculated; farther away, the reflux area gradually decreased and the velocity and turbulence distribution trended towards uniform. Secondary flows occurred in different directions, which improved the diffusion of materials and DO retention. The system is effectively self-circulating, and the plug-flow capability may be scaled up for commercial application.

# 1. Introduction

The rapid development of China's economy has been accompanied by an increasing demand for food sources near urban areas. To meet this challenge, China has become the world's largest producer and consumer of aquaculture products [1,2]. Intensive aquaculture is an important strategy for addressing the dual shortages of farm and water resources. As currently practised in urban China, intensive aquaculture has suffered a series of problems, such as high ammonia and nitrogen concentrations [3], excessive levels of water consumption, aquatic pollution [4] and recurrent fish diseases [5]. The water in which aquaculture is carried out serves a range of functions: it acts as a habitat for fish, a site for the decomposition of faecal matter and bait and a location for the growth of plankton. Overall, therefore, the functional division between consumer, producer and decomposer in aquaculture ponds remains unclear, presenting challenges to the management of ponds and inevitably resulting in ecological imbalance [6]. Advances in aquaculture practice are therefore required to ensure that the ecological functional zones remain independent. The delineation of these functions should enable improvements in fish production and water quality.

In 2005, Chappell, in collaboration with a local farmer, invented an in-pond raceway system (IPRS) at Auburn University [7]; this system is a simplified version of the partitioned aquaculture system (PAS) that originated at Clemson University [8,9]. IPRS consists of several aquaculture raceways installed in a pond, with separate breeding and purification areas. The raceways maintain the captivity of the feeding fish, and the purification area is inhabited by filter-feeding fishes and other aquatic organisms [7–10]. Using this method, it is easier to control fish excrement disposal during the aquaculture period. The addition of pumps and waste collection equipment allows excreta and residual feed to accumulate in the downstream purification area for easier collection and treatment. Although the initial purpose of the IPRS was to improve the catch rates [10,11], this system is also useful for countries or regions with high land use costs or other environmental pressures [9,12]. The USA has 12 large aquaculture fields that use IPRS [13]. In 2013, Zhou, who had been working in the United States Soybean Association, introduced IPRS to China, where it was implemented in Jiangsu, Anhui, Shanghai and several other locations [14]. IPRS has since been extended to more than 18 provinces and cities, with an aquaculture footprint of 200 000 $m^2$ in China.

Water flow is an important environmental factor that affects the behaviour of fish. It stimulates various reactions among fish, which affects their growth [15–18] and may generate behavioural variability within a species, generally in response to velocity distribution and turbulence [19]. Turbulence plays a critical role in the transport and dispersal of excreta, nutrients and pollutants [20,21]. Dissolved oxygen (DO) is another critical factor to consider in aquaculture ponds. When DO concentrations in a water body drop below $5 \, mg \, l^{-1}$, fish appear sluggish and display a behaviour known as 'floating head'. They do not respond to bait and may even suffocate, resulting in significant economic losses [22–24]. In IPRS, one or more high-power pumps are used to pump water into the raceways, forcing the water to flow out the raceways and into the purification area, thereby circulating it and forming a pathway for fish. The flow patterns in the original aquaculture raceways are similar to the plug-flow in pipes, with uniform velocity distribution in cross-sections [25]. However, despite improvements in the flow characteristics to increase mixing, the term 'plug-flow' is still used to describe the flow of water being pumped downstream in aquaculture raceways. A rectangular raceway occupies 20% less space and costs less than a circular tank; however, there is little mixing, causing water quality and DO to vary significantly from inlet to outlet [26,27]. After several additional innovations were made in the design and operational procedures, the IPRS water exchange was improved and DO concentrations increased.

An IPRS (figure 1) with an aeration plug-flow device (APFD) was designed by farmers in China. The APFD consists primarily of a curved baffle and a set of micro-bubble tubes. It is fixed on the inlet of the raceway and the micro-bubble tubes are submerged at a specific depth, so that the air from the external compressor is pumped into the tubes, causing micro-bubbles to form. The floating micro-bubbles are blocked by the top of the curved baffle and forced to change their original path, which causes the water current to flow from the bottom to the top, from the raceway inlet to the outlet and finally into the purification area. This forms a recirculating flow in the IPRS, and mass transfer of oxygen occurs between the interface of the micro-bubbles and the water, resulting in increased DO concentration in the water. The IPRS with APFD enables water recirculation and adds DO to the aquaculture pond simultaneously. These features make this system increasingly popular in China.

However, there is a lack of sufficient research exploring the effects of APFD on the hydrodynamic characteristics in raceways and the improvements in plug-flow resulting from the use of APFD in IPRS. Farmers often implement IPRS with APFD without first considering the specific hydrodynamic

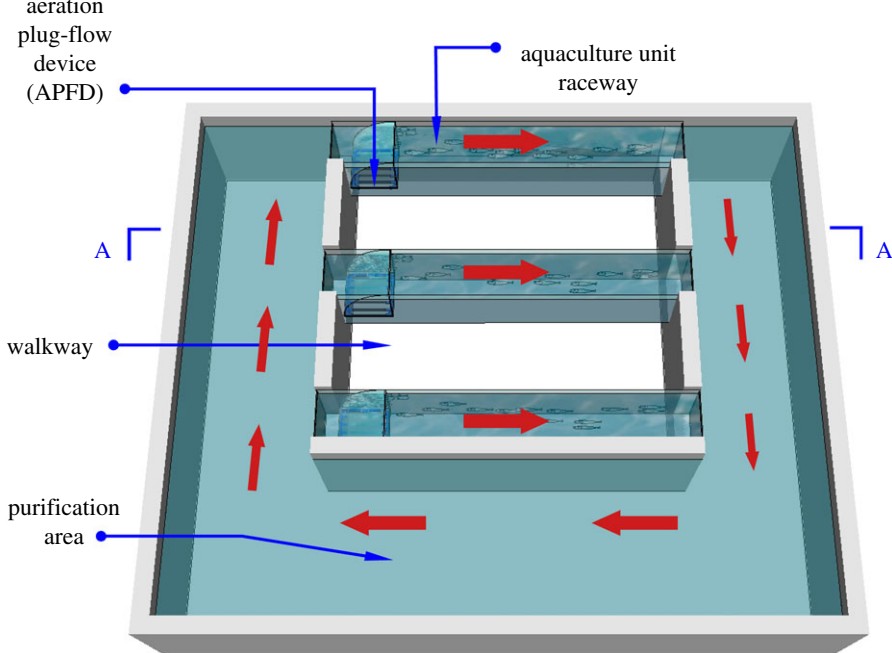

**Figure 1.** Diagram of the IPRS with APFD.

conditions of the aquaculture pond. They are confused about the hydrodynamic conditions present in raceways under a specific air flow rate and how to set up or improve APFD. At some sites, the water recirculation velocities or turbulence exceed the preferred conditions for fish, which may compromise fish production and cause economic losses. The full potential of the APFD innovation cannot be realized without a better understanding of its underlying hydraulic characteristics. In this study, we aimed to elucidate the effect of APFD on hydrodynamics in raceways in aquaculture ponds. To achieve this, we designed an indoor physical model of an IPRS with APFD and regulated the aeration flow rate. We performed a series of velocity measurements in the raceway, analysing the distribution and turbulence and calculated the plug-flow characteristics of the APFD. Our findings should provide a reference for future-optimized design of the IPRS and thus promote intensive aquaculture.

# 2. Experimental set-up and methods

## 2.1. Aeration plug-flow device

The detailed structure of the APFD, as shown in figure 2, consisted of a framework (1), an aeration device (2) and an arc-shaped baffle (3). The aeration device included a group of gas pipes (4) and a group of micro-bubble-generating tubes (5). One of the pipes in the gas pipes group was composed of an air inlet (6) and a gas transmission pipe (7). The group of micro-bubble-generating tubes included tee pipes or adapters (8), five micro-porous aeration tubes (9) and gas nozzles. The gas nozzles were connected to the tee pipes or adapters through an internal thread.

The width of the APFD was set to be the same as that of the raceway. The APFD was fixed at the inlet and near the bottom of the raceway, and air pumped by an external air compressor was passed through the inlet hole (6 in figure 2), eventually reaching the micro-porous aeration tubes (9 in figure 2). This generated a large number of tiny bubbles that floated from the bottom upwards.

The actual appearance of the micro-porous aeration tubes is shown in figure 3. They are made of a nano-polymer material with stable physical and chemical properties and have outer and inner diameters of 15 mm and 10 mm, respectively. The surface of the aeration tube has several tiny holes, with a density between 700 and 1200 holes per metre length. The average diameter of these holes is 50 μm, which enables the generation of micro-bubbles that have a longer duration of residence and more complete contact with water than large bubbles. Once the air is pumped into the aeration tube, increased air pressure opens the aeration orifices; otherwise, they are shut off automatically due to water pressure.

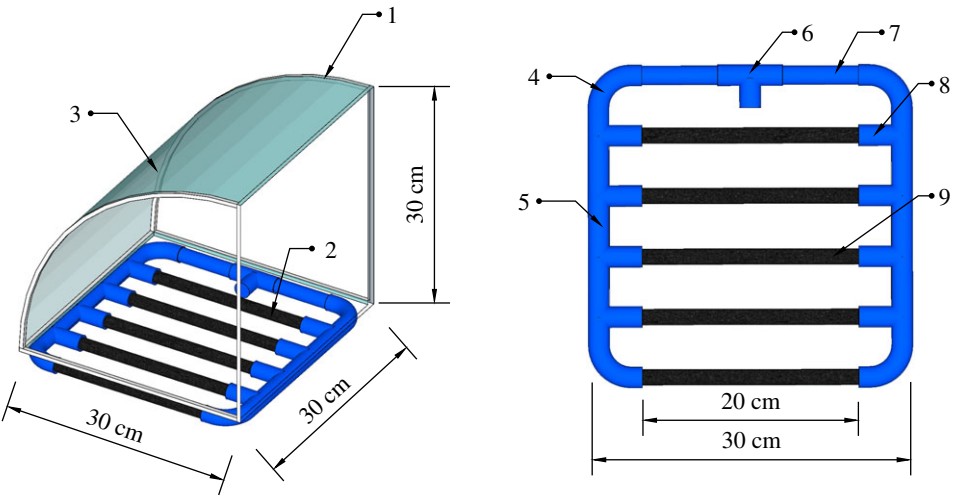

**Figure 2.** Detailed structure of the aeration plug-flow device. 1: framework; 2: aeration device; 3: arc-shaped baffle plate; 4: group of gas pipes; 5: group of micro-bubble-generating tubes; 6: air inlet; 7: gas transmission pipe; 8: tee pipe or adapter; 9: micro-porous aeration tubes.

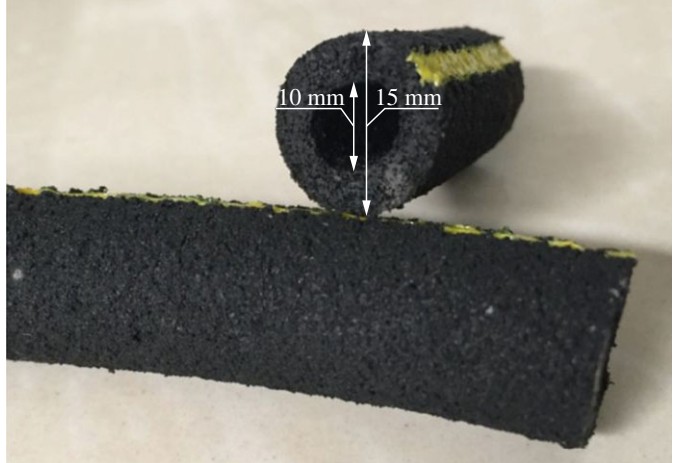

**Figure 3.** The micro-porous aeration tube (Photo: Hu).

This feature effectively avoids clogging by sediments or dust. The air bubbles produced from this aeration tube are small enough to enhance the total area of contact with the water.

## 2.2. The indoor in-pond raceway system

Our indoor IPRS model consisted of three APFDs, three aquaculture raceways and a purification area, as shown in figure 1. The four sides and the bottom of the model were reinforced with 10 cm of concrete. The overall size was 370 cm × 310 cm × 75 cm (length × width × height), with the concrete wall higher than the water level, ensuring the water could not overflow. A walkway was installed to facilitate experimental operation in between the two raceways.

A profile of the raceway is shown in figure 4a. The size of each raceway was 220 cm × 30 cm × 40 cm (length × width × height). Each raceway was made of plexiglass, and the entrance and exit of the raceway both connected to the purification area (figure 1). The APFD drove the circulatory system and was installed at the front of the raceway, 10 cm from the bottom.

After the air compressor was turned on and the pressure and air flow rate were adjusted, the micro-porous aeration tube continuously produced a large number of floating micro-bubbles from the bottom. When the bubbles are subjected to the obstruction of the upper arc-shaped baffle, they are forced to change their original direction, flowing out from the front of the curved baffle and forcing other floating bubbles to also change their path (shown in figure 4b). The APFD causes the air bubbles to

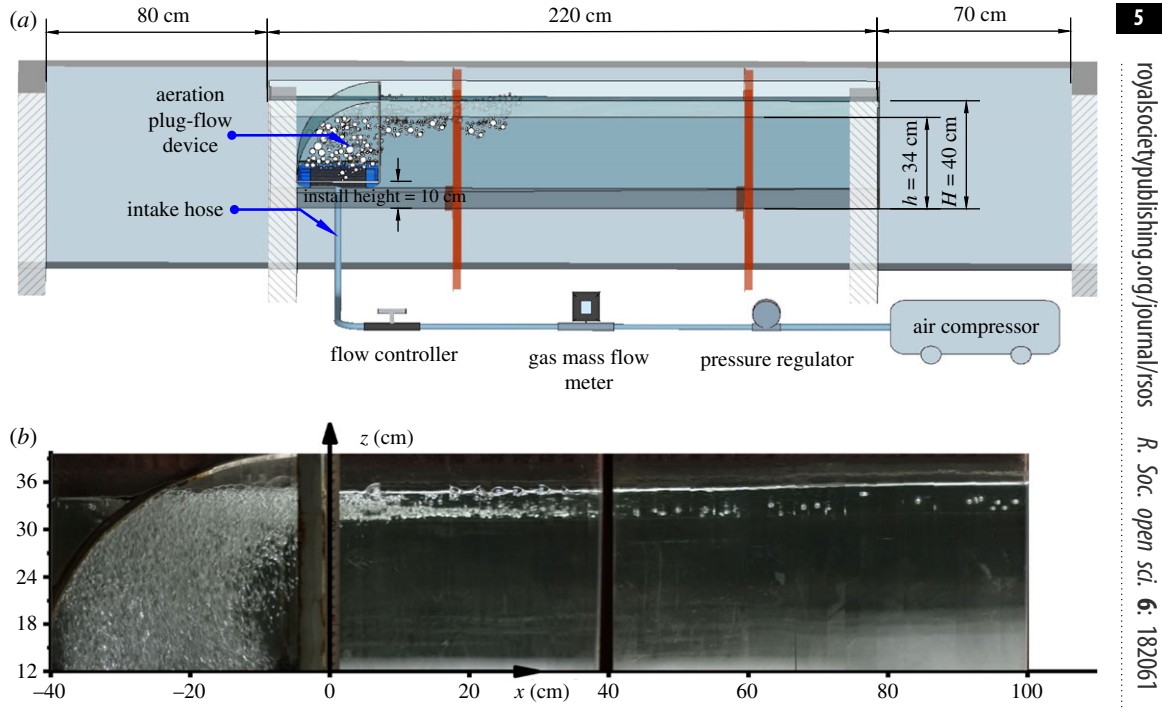

**Figure 4.** (a) Equipment layout and sizes of aquaculture unit raceway (A–A profile of figure 1). (b) Photograph showing the movement of the bubble (Photo: Hu).

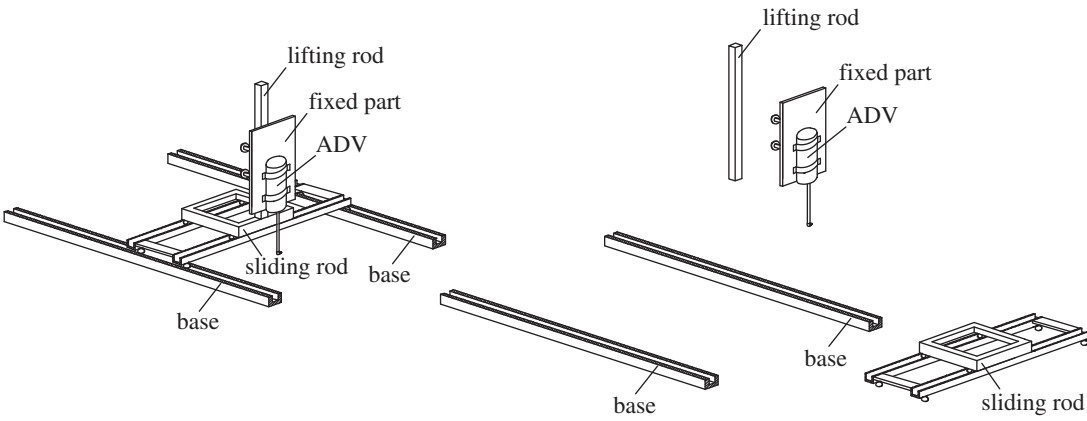

**Figure 5.** Diagram of the displacement device.

flow from the bottom to the surface and from the inlet to the outlet of the raceway, which drives the water to move through the aquaculture raceway, passing through the purification area, and returning to the inlet. As indicated by arrows in figure 1, the water flow of the entire system circulates in a clockwise direction, simplifying operation and monitoring.

## 2.3. Flow velocity measurement

This research focused on the effect of the APFD on the hydrodynamics of the IPRS. Therefore, fish were not present in the raceway as they may have interfered with the hydrodynamics of the system and confounded the results. Furthermore, the three parallel raceways had similar flow fields; therefore, only one of them was chosen for measurement.

Measurements of the three-dimensional velocities were conducted via a Nortek Acoustic Doppler Velocimeter (ADV), and a custom displacement device was designed. The displacement device consisted of a base, a sliding rod, a lifting rod and a fixed part (figure 5). The ADV was mounted on the fixed part and allowed to slide freely along the longitudinal, transversal and vertical directions of the raceway to measure any point in the water.

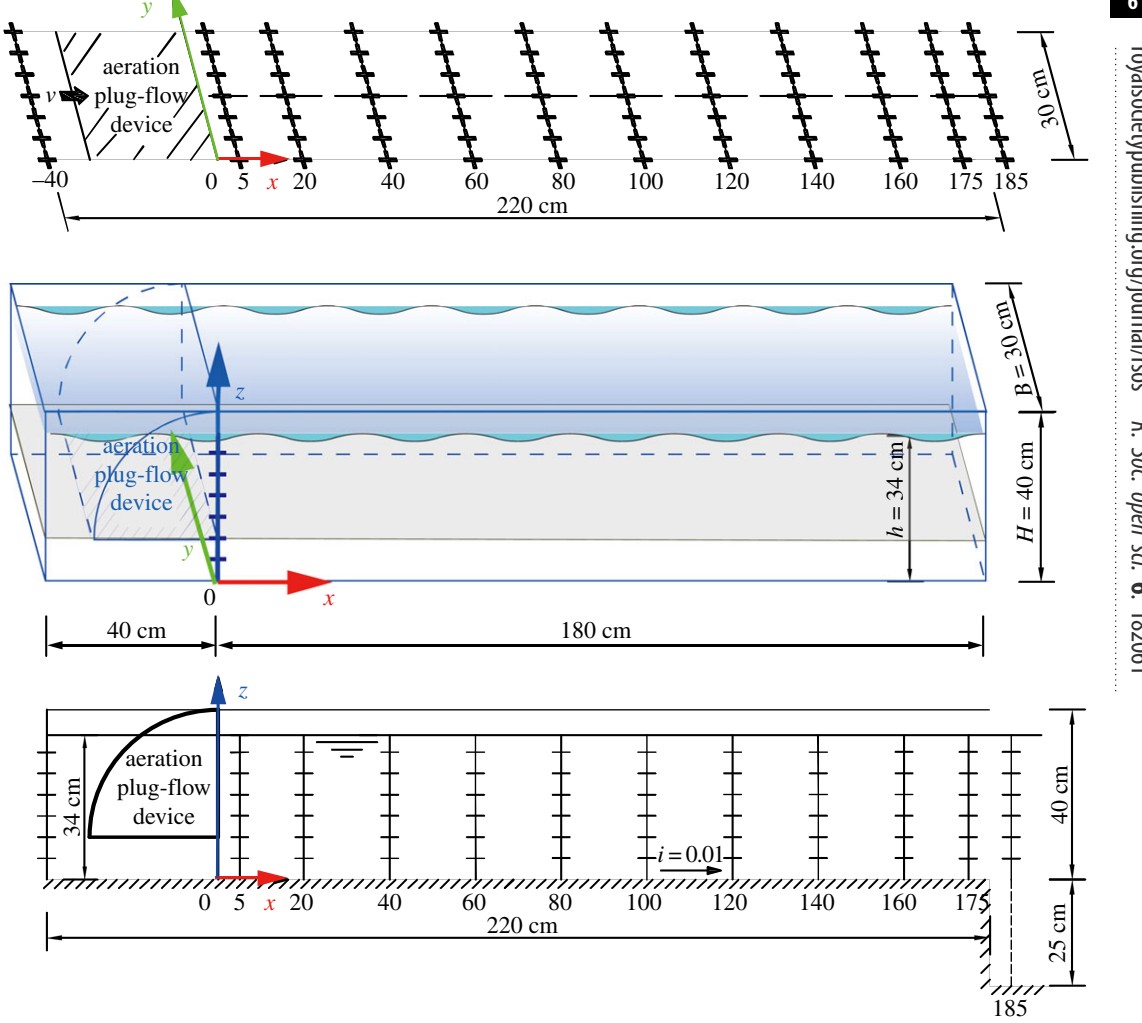

**Figure 6.** Diagram showing the coordinate system and arrangement of measured points.

**Table 1.** Measurement sets and reference division.

| height from bottom $z$(cm) | relative height $z/h$ (water depth $h = 34$ cm) | Nezu and Nakagawa's division of the water in open-channel shear flow | |
|---|---|---|---|
| 5 | 0.15 | $z/h = 0 - 0.15$ | wall region |
| 10 | 0.29 | $z/h = 0.15 - 0.6$ | intermediate region |
| 15 | 0.44 | | |
| 20 | 0.59 | | |
| 25 | 0.74 | $z/h = 0.6 - 1.0$ | free surface region |
| 30 | 0.88 | | |

The direction of the ADV probe affected the Reynolds shear stress and the secondary flow; therefore, to minimize error from the probe, the positive direction of the probe was aligned with the $x$-axis, as shown in figure 6. The $x$-direction was parallel to the raceway, the $z$-direction was vertical along the ADV rod, and the $y$-direction was set at a $90°$ angle to the $x$-direction.

The distances from the bottom ($z = 5$, 10, 15, 20 , 25 and 30 cm) were measured in accordance with Nezu & Nakagawa's [28,29] division of water flow in an open-channel into three regions and our particular experimental conditions. The relative heights were 0.15, 0.29, 0.44, 0.59, 0.74 and 0.88, as shown in table 1.

For each set of measurements, ADV data were sampled at a frequency of 25 Hz within a regular grid consisting of 12 laterals of 6 points 5 cm apart, producing a grid of 72 points (figure 6). The signal-to-noise ratio (SNR) was maintained at 15 or above. ADV sampling was performed for a minimum of 60 s at each point, with an average sampling time of 100 s.

## 2.4. Data filtering and processing

To ensure the authenticity of the measured data, we excluded instances where the correlation coefficient was less than 80%. This left approximately 2000 measurements at each point.

Instantaneous three-dimensional flow velocity components were acquired at each measurement point using the ADV. Measurements were made in the longitudinal, lateral and vertical directions, denoted as $u_i$, $v_i$ and $w_i$. The instantaneous velocity can be divided into the mean velocity and the fluctuating component, as shown in the following equations

$$u_i = u + u'_i \qquad (2.1a)$$

and

$$u'_i = u_i - u, \qquad (2.1b)$$

where $u_i$ is the instantaneous velocity ($\text{cm s}^{-1}$), $u$ is the mean velocity ($\text{cm s}^{-1}$) and $u'_i$ is the fluctuating velocity ($\text{cm s}^{-1}$). The mean velocity $u$ can be obtained by

$$u = \frac{1}{N} \sum_i^N u_i, \qquad (2.2)$$

where $N$ is the number of valid data points. The strength of the turbulence in the longitudinal velocity $u'$ ($\text{cm}^2\text{s}^{-2}$) can be defined as:

$$u' = \sqrt{\overline{u_i'^2}} = \sqrt{\frac{1}{N} \sum_i^N (u'_i)^2} = \sqrt{\frac{1}{N} \sum_i^N (u_i - u)^2}. \qquad (2.3)$$

Similar definitions can be applied to the lateral and vertical velocities $v_i$ and $w_i$.

The turbulent shear stress (i.e. Reynolds shear stress) is calculated as:

$$\tau_{uv} = |u'v'|, \ \tau_{uw} = |u'w'|, \ \tau_{vw} = |v'w'|, \qquad (2.4)$$

where $u'v'$, $u'w'$ and $v'w'$ are the covariance of instantaneous fluctuating velocity.

The turbulent kinetic energy $k$ ($\text{cm}^2\text{s}^{-2}$) is:

$$k = \frac{1}{2}(u'^2 + v'^2 + w'^2). \qquad (2.5)$$

# 3. Results and discussion

To investigate velocity distribution and turbulence in the raceway, the patterns of mean velocity, Reynolds shear stress and turbulent kinetic energy were derived from the ADV measured data. The APFD induced water flow from the bottom to the surface and from the inlet to the outlet of the raceway; therefore, we were primarily concerned with the velocity patterns in the horizontal sections and the cross-sections.

## 3.1. Velocity patterns in horizontal sections

A schematic diagram of various parts of the horizontal section is shown in figure 7a, where the sections of $x = -40$ cm and $x = 180$ cm represent the inlet and the outlet. For convenience, $x = 0-120$ cm was defined as the fore-mid part and $x = 120-185$ cm was defined as the rear part.

The velocity patterns in the horizontal sections were plotted, where (a)–(f) represented the velocity distributions at heights $z = 5, 10, 15, 25$ and 30 cm. The velocities at the baffle were set to zero. The colour indicates the velocity value of $u + v$, the blue vector arrows indicate the direction of $u + v$, the grey dotted lines denote the isoline of $u + v = 0$ and the shaded part denotes the position of the APFD.

As shown in figure 7b, the flow in the raceway was divided into plug-flow ($u + v > 0$) and reflux ($u + v < 0$); the grey dotted lines were the boundary lines. The range of the reflux and the area proportions

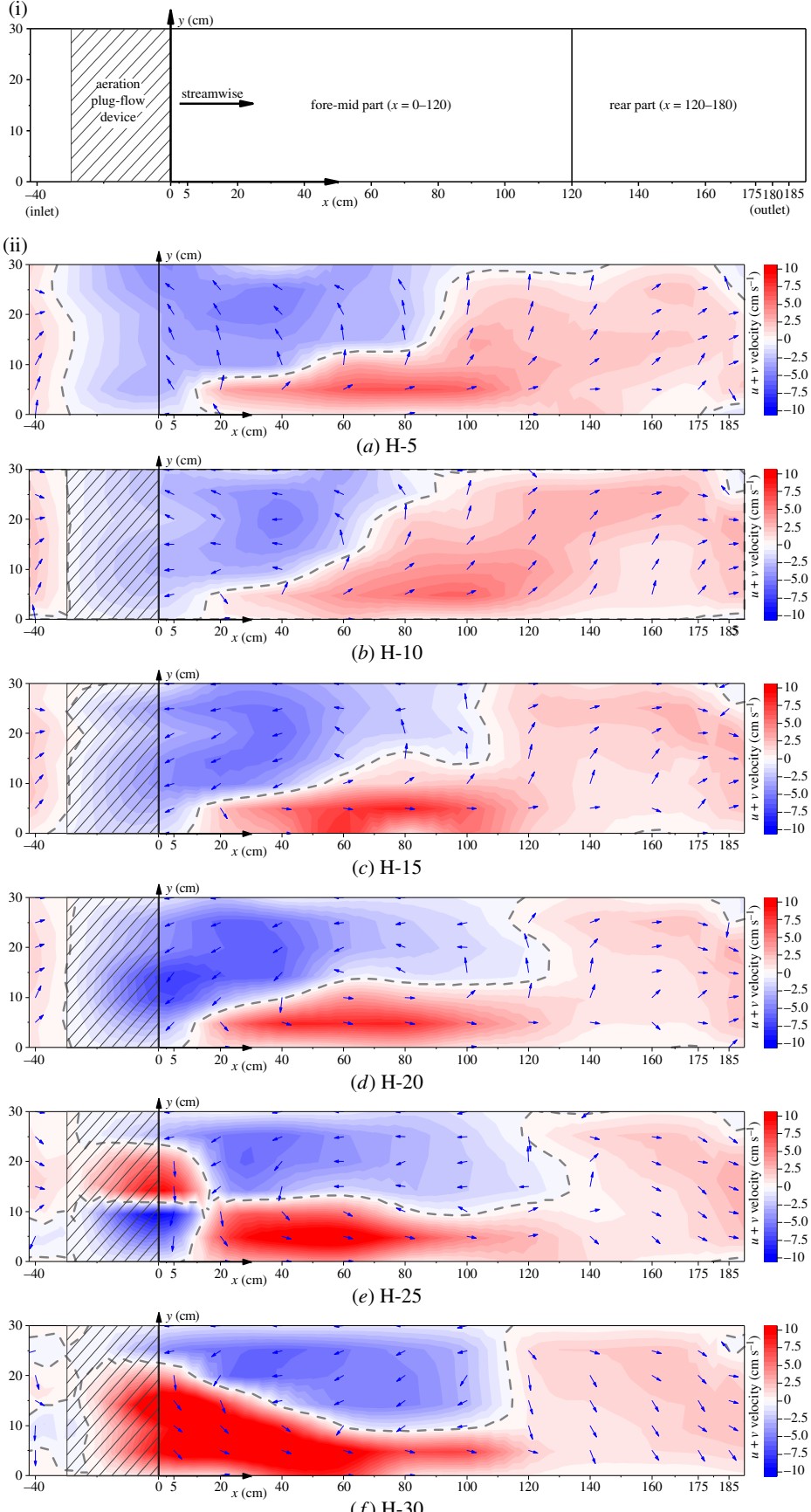

**Figure 7.** (i) Schematic diagram of various parts of the horizontal section. (ii) Velocity patterns in the horizontal sections. Colour represents the horizontal patterns of the velocity. Vectors describe the direction of the horizontal velocity $(u, v)$. The grey dotted lines denote the isoline of $u + v = 0$.

**Table 2.** The range of reflux and the ratio between plug-flow and reflux.

| height from bottom $z$ (cm) | range of reflux (cm) | plug-flow proportion (%) | reflux proportion (%) |
|---|---|---|---|
| 5 ($z/h = 0.15$) | 0–91 | 0.64 | 0.36 |
| 10 ($z/h = 0.29$) | 0–77 | 0.68 | 0.32 |
| 15 ($z/h = 0.44$) | 0–99 | 0.62 | 0.38 |
| 20 ($z/h = 0.59$) | 0–114 | 0.57 | 0.43 |
| 25 ($z/h = 0.74$) | 0–120 | 0.58 | 0.42 |
| 30 ($z/h = 0.88$) | 0–108 | 0.65 | 0.35 |
| **average** | 0–102 | 0.62 | 0.38 |

between the plug-flow and the reflux are listed in table 2. The range of the reflux extended to approximately $x = 100$, and the area ratio was approximately 6 : 4. In the plug-flow area, the water was pushed through the raceway for self-circulation. The reflux contributed to the retention of the DO in the aquaculture unit for consumption by fish.

At each height, the patterns at the fore-mid part ($x = 0$–120 cm) and the rear part ($x = 120$–180 cm) were distinct (figure 7). The plug-flow and the reflux interacted at the fore-mid part, where the velocity was generally higher than at the rear part. There were two cores of high velocities at the fore-mid part, encircled by diminishing velocities and separated on two sides of the raceway. The two cores formed by the differential effects were the red core formed by the plug-flow, and the blue core formed by the reflux. At the rear part, the velocity was relatively uniform, mostly reflecting the plug-flow effect, and the strength did not vary much with height. The velocities at the fore-mid part varied obviously with height, and the changes were mainly reflected in the movement of the high-velocity cores, the magnitude of the velocity and the reflux range.

Near the bottom ($z = 5$ cm), the reflux zone was near the APFD, while the plug-flow zone was slightly behind. The plug-flow effect generally began at $x = 10$ cm. When approaching the water surface (i.e. at $z = 25$ cm) another plug-flow zone appeared at the position of the APFD. Until $z = 30$ cm, the two plug-flow zones developed and merged, showing the tendency of the plug-flow zone to form next to the APFD, and the reflux zone to form farther from it. The velocity within the plug-flow and the reflux regions at the fore-mid part gradually increased; however, the increase in the velocity of reflux was less than that for the plug-flow. The reflux range at $z = 10$ cm was the shortest among the measured heights (table 2), even shorter than the range at $z = 5$ cm, which was lower than the installation height of the APFD. Above the installation height, the reflux range gradually increased at heights of $z = 15$–25 cm and reduced to approximately $x = 100$ cm at $z = 30$ cm. At heights of $z = 10$ and 30 cm, the area ratio between the plug-flow and the reflux was approximately 7 : 3.

The blue vectors represent the patterns of secondary flows that occurred at each height (figure 7). The secondary flow patterns were divided into two parts, and their rotation directions were opposite, i.e. at the fore-mid part, the water flow rotated counter-clockwise, whereas at the rear part, it rotated clockwise. The vectors show that as the height increased from the bottom to the surface, the secondary flow cell at the fore-mid part moved backward and the curvature of the vector at the rear part increased.

The velocities at the inlet and outlet of the raceway show that most of the $u + v$ values were greater than zero, indicating that under the influence of APFD, the water was effectively pushed through the raceway, thus flowing into the purification zone and returning to the aquaculture unit for self-circulation.

## 3.2. Velocity patterns in cross-sections

The velocity distributions in each cross-section are shown in figure 8. The colour shows the velocity value of $v + w$, with the blue vector arrows indicating the direction of $v + w$. The grey dotted lines represent boundaries between the plug-flow and the reflux, which were consistent with the horizontal velocity patterns above. In reference to figure 7b, the plug-flow and the reflux zone are distinguished and separated on two sides of the dotted line.

At the inlet and the rear part, i.e. C-(40) and C-140–185, the flow was mostly pushed and the velocities were uniform. At the fore-mid part (C-5–120), there was a plug-flow and a reflux zone.

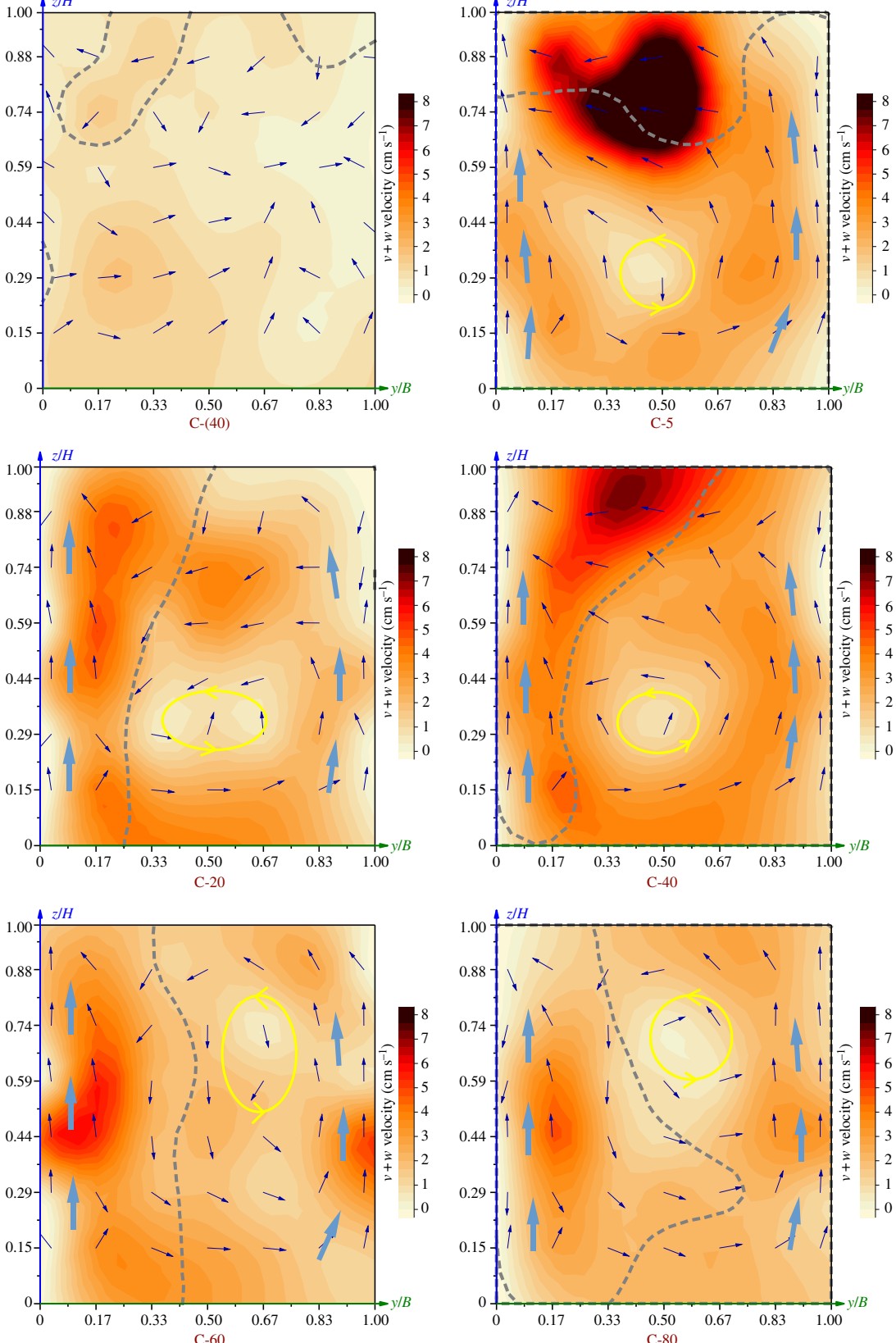

**Figure 8.** Velocity patterns in the cross-sections. Colour represents the cross-sectional velocity patterns. Vectors indicate the direction of the cross-sectional velocity ($v$, $w$). The grey dotted lines denote the isoline of $v + w = 0$.

Velocities varied greatly with the cross-sections and were mostly larger on the plug-flow side. In the two cross-sections immediately downstream of the APFD, i.e. C-5 and C-20, there were high-velocity cores that were primarily distributed in the plug-flow zone. The low-velocity cores were located in the

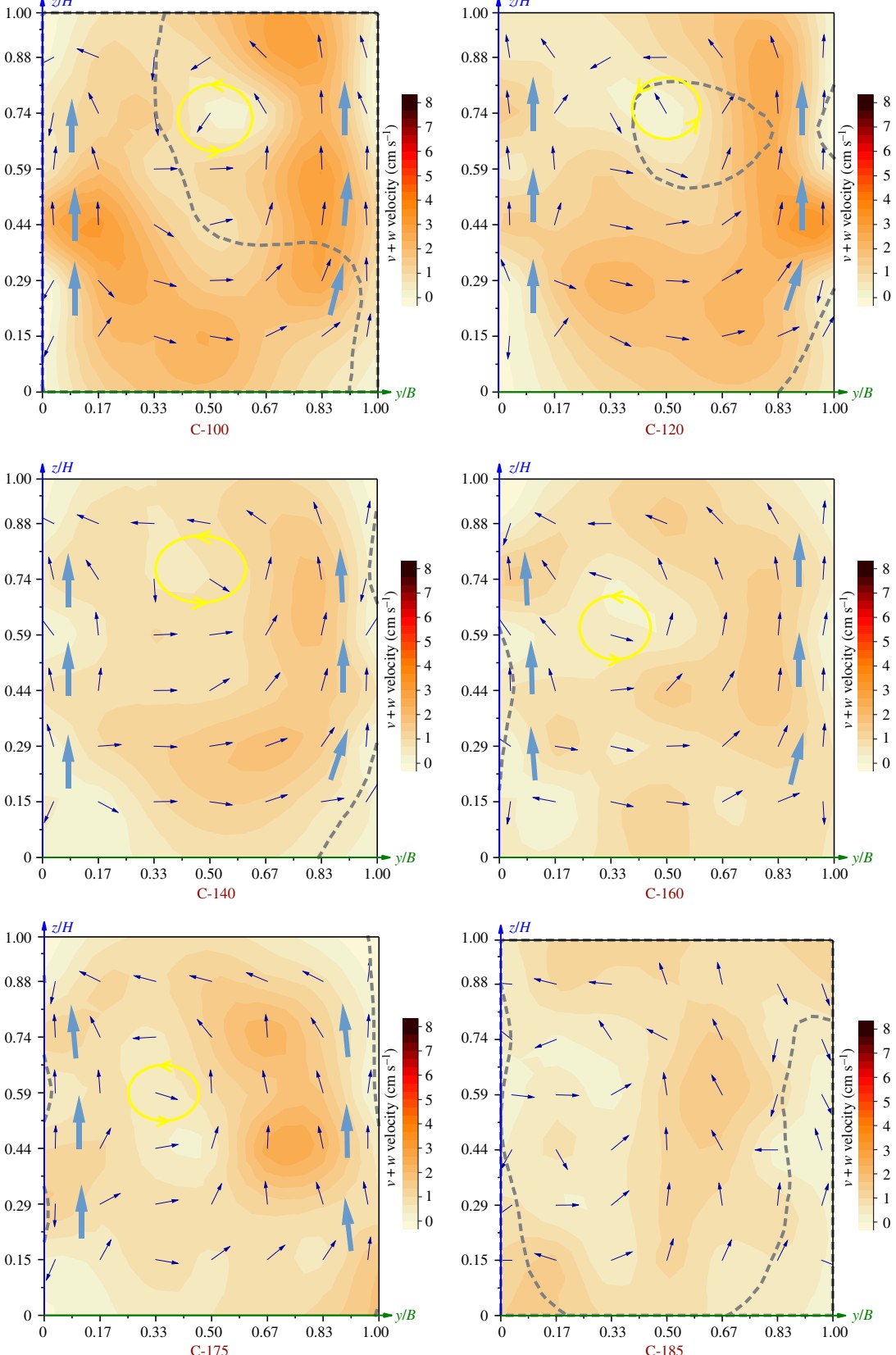

**Figure 8.** (*Continued.*)

reflux zone. The plug-flow area was smaller than the reflux area in these two sections. At the following sections, i.e. C-40–120, the plug-flow area gradually expanded. There were low-velocity cores distributed in the reflux zone, and the velocities of the entire section tended to be uniform.

The vectors show that there were secondary flows at the centre, and the flow near the side walls of the raceway moved from the bottom to the surface (i.e. the large blue arrows in figure 8). The secondary flow at the centre rotated counter-clockwise and its cell approximately coincided with the low-velocity core in the reflux zone (as shown by the yellow circle in figure 8).

## 3.3. Turbulence characteristics

In the self-aeration models with high water heads and flow rates, air bubbles play an important role in drag reduction and accelerate the rate of dissipation of turbulent kinetic energy [30]. However, in this study, the movement of air bubbles drove the circulation of the water in the system. The existence of air bubbles should be positively correlated with turbulence.

In figure 9, the lines with scattered points represent the Reynolds shear stress in each direction, and the colour indicates the magnitude of the turbulent kinetic energy. The sub-graphs (a)–(f) represent the distribution of turbulence characteristics at heights $z = 5, 10, 15, 25$ and $30$ cm.

At each height, the turbulence characteristics were not evident at the inlet ($x = -40$ cm) and the rear part ($x = 140–185$ cm) but appeared intensively at the fore-mid part ($x = 0–120$ cm) of the raceway. Combined with figure 7$b$, this shows that both the Reynolds shear stress ($\tau$) and the turbulent kinetic energy ($k$) in the plug-flow zone were higher than in the reflux zone. The values of these turbulence characteristics gradually increased near the water surface.

The patterns of the Reynolds shear stress ($\tau$) in the three directions were similar. Besides the maximum value in the plug-flow zone, there was a smaller peak in the reflux zone. The Reynolds shear stress in each section was compared to show that the value of $|u'w'|$ was larger than in the other two directions, while the values of $|u'v'|$ and $|v'w'|$ varied little, indicating that the turbulence in the longitudinal and vertical directions was relatively large and the strength did not differ much.

The turbulent kinetic energy ($k$) showed a single trend with height. Only one core of high value was located in the plug-flow zone. Close to the water surface, $k$ increased to a maximum and the core tended to move to the APFD. This movement trend was the same as the movement of the high-velocity core of plug-flow with height (figure 7), showing that the velocity at the fore-mid part was correlated with turbulence.

The experimental fluid was a type of bubble-induced multiphase flow in the circulating system. The movement of the bubbles induced turbulence, and the centripetal force was generated by the circulation system. Secondary flows in the different sections were evident. According to Prandtl [31], secondary flow can be divided into two categories: the first kind is the secondary flow caused by centripetal force, which is common in river bends; the second kind is the secondary flow that can be observed in some straight channels because of turbulence anisotropy [32]. The mechanism underlying secondary flow of the second kind was first deduced by Einstein & Li [33] on the basis of Reynolds Average Navier–Stokes (RANS) equations; numerous numerical simulations have since been performed in an attempt to reproduce it [32,34]. Flow turbulence was more intense at the fore-mid part of our experimental system, and the secondary flow of Prandtl's second kind was obvious, as shown in figures 7 and 9. The turbulence at the rear part was weaker and the centripetal force made a greater contribution to the water flow, primarily generating secondary flows of Prandtl's first kind. Previous studies have found that secondary flow influences the suspended sediment concentration [35], affects the near-bed pattern to some extent [36] and promotes transport processes [37]. Secondary flow and turbulence can prevent the precipitation of residual bait and faeces in raceways and promote the mixing of materials and energy, helping to maximize their utilization by fish.

The patterns at heights of $z = 5$ cm and $z = 10$ cm are worthy of attention. The height of $z = 5$ cm was located below the installation height of the APFD and the $z = 10$ cm height was the APFD installation height where the air bubbles formed and floated; therefore, these two heights were little affected by bubbles. Nevertheless, patterns similar to those in the upper heights were observed at these two heights. This was attributed to the presence of secondary flows in the cross-sections (figure 8); the turbulence was transmitted in the vertical direction and obvious turbulence remained even in the wall region (in reference to the division of table 1).

## 3.4. Plug-flow characteristics

The velocities in the last three cross-sections ($x = 160–185$ cm) were mostly positive, and there was little reflux (figure 8). To measure its cycling efficiency, the average flow rate of the three sections was used to represent the overall flow rate of the raceway. The effects of the lateral and vertical velocities were ignored ($v$ and $w$; accordingly, the plug-flow rate was calculated using the streamwise velocity ($u$)

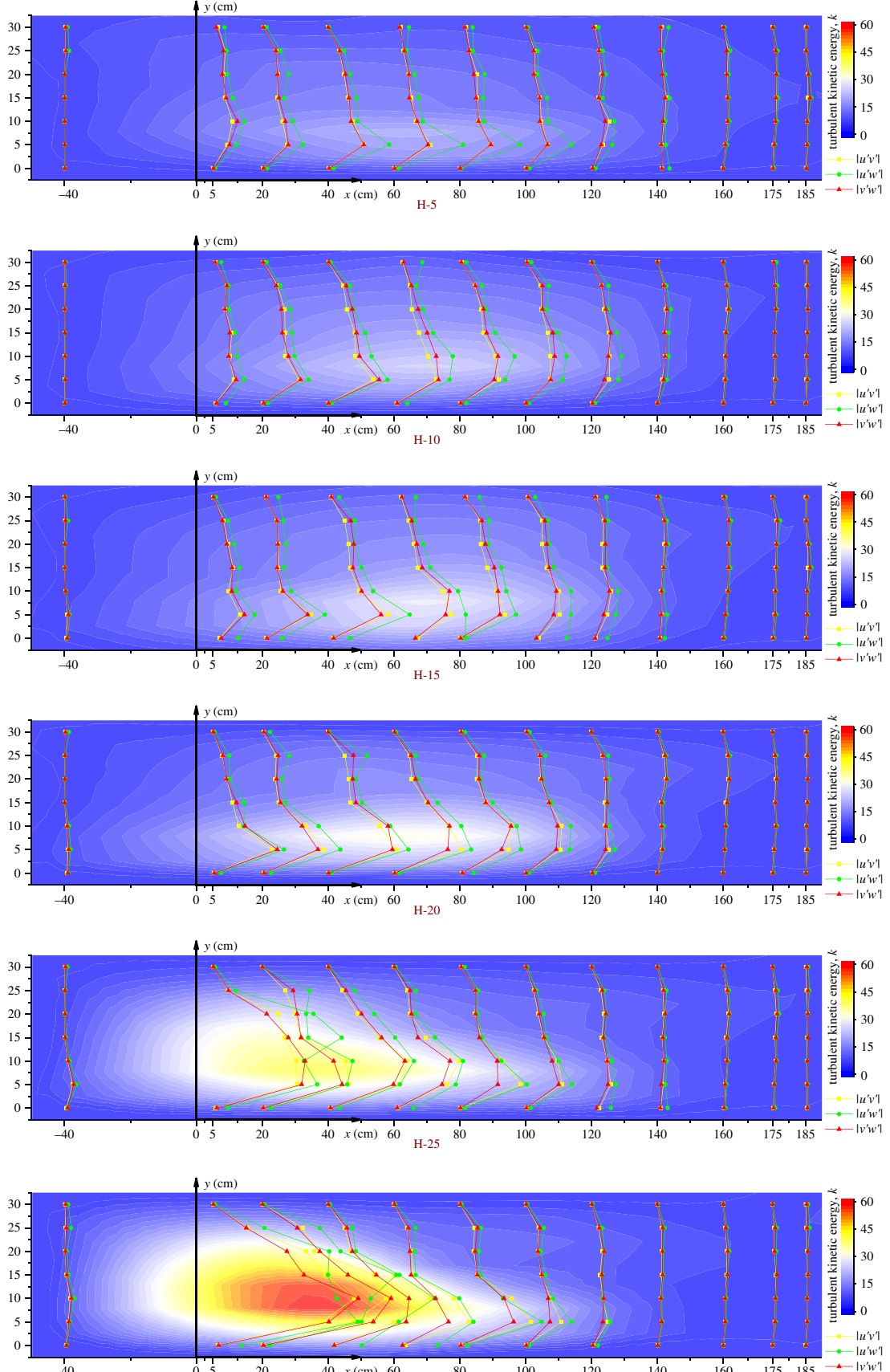

**Figure 9.** Turbulence characteristics in the horizontal sections. Colour represents the horizontal patterns of the turbulent kinetic energy. Lines indicate the Reynolds shear stress in the three directions.

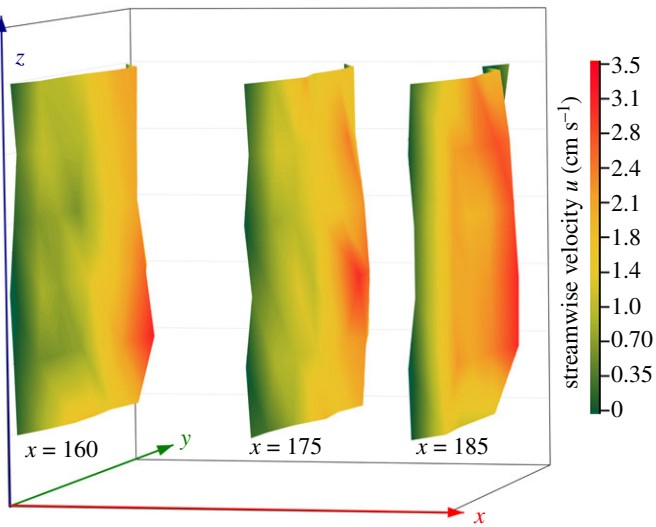

**Figure 10.** The integration of the last three cross-sections.

**Table 3.** Integral flow rate of the last three cross-sections.

| cross-section (cm) | integral flow rate of each section (cm³ s⁻¹) | average flow |
|---|---|---|
| $x = 160$ | 902.8 | $Q = 994$ cm³ s⁻¹ |
| $x = 175$ | 994.1 | |
| $x = 185$ | 1087.0 | |

according to the following equation

$$Q_{160} = \iint u\,(160,y,z)\,\mathrm{d}y\mathrm{d}z \tag{3.1}$$

where $Q_{160}$ (cm³ s⁻¹) is the flow rate at cross-section $x = 160$, and $u(160,y,z)$ is the streamwise velocity at each point of measurement point at $x = 160$. The flow rates at cross-sections $x = 175$ cm and $x = 185$ cm (i.e. $Q_{175}$ and $Q_{185}$) were calculated similarly. The plug-flow rates of the last three sections are shown in figure 10. The computed results for the flow rate at each cross-section are shown in table 3. The calculated average flow was 994 cm³s⁻¹ and the average streamwise velocity was $U = 0.97$ cm s⁻¹.

The specific flow pattern in the raceway was ignored; therefore, the cycling efficiency can be represented by the average flow rate. The water retention time was calculated by the following equation

$$t = \frac{V}{Q}, \tag{3.2}$$

where $V$ (cm³) indicates the volume of water in the raceway. Retention time was estimated as $t \approx 226$ s $\approx 3.8$ min. There was reflux in the raceway; therefore, the retention time was likely longer, and the calculated value underestimated it. This also meant that oxygen remained in the raceway for a long duration rather than being simply transferred downstream by the flow, thus increasing the efficiency of aeration.

The IPRS is applied to commercial-scale operations in the USA and China. The general engineered size of the IPRS in China is 20–25 m long, 4–5 m wide and 1.5–2 m deep. The longitudinal and lateral linear scales between the prototype and the model were $\lambda_L = 5/0.3 = 16.7$, and the vertical linear scale was $\lambda_V = 2/0.34 = 5.88$. To extend the experimental results into the field according to similarity principles, the scale of the average velocity was $\lambda_v = \lambda_V^{0.5} = 2.4$, and the scale of the average flow rate was $\lambda_Q = \lambda_L \lambda_V^{1.5} = 238$. The average velocity was about $U_P = 0.024$ ms⁻¹ and the average flow rate in the prototype reached $Q_P = 0.237$ m³s⁻¹ $= 853$ m³h⁻¹. The length of each raceway primarily influences the breeding capacity and density of the system, which has a small influence on the flow rate and exchange time; therefore, it is rarely considered in the design. The longitudinal scale was ignored, and we attempted to adapt the experimental model for the commercial-scale application, as described by

Brown *et al.* [12]. The size of each raceway in Brown's experiment was 7.71 m × 4.88 m × 1.22 m (length × width × depth). The lateral linear scale between the two models was $\lambda_L = 4.88/0.3 = 16.3$, and the vertical linear scale was $\lambda_V = 1.22/0.34 = 3.6$. The scale of flow rate was $\lambda_Q = \lambda_L\lambda_V^{1.5} = 111$. The conversion flow rate was $0.11 \text{ m}^3 \text{ s}^{-1}$, which was close to Brown's experimental result of $0.15 \text{ m}^3 \text{ s}^{-1}$. It shows that the APFD in the present experiment can effectively stimulate water flow at a commercial scale.

This study advances the knowledge of the hydrodynamic characteristics of an IPRS with APFD. The results provide test data for further numerical simulations and guidance for field trials to optimize the design. However, there are potential limitations in this study that should be pointed out. The experiment was performed without fish, as they would affect the hydrodynamics and water quality in the raceway. The conditions in turn affect the physiological function and behaviour of the fish. Their movements increase turbulence, which affects the hydrodynamic conditions such as the drag coefficient, the average velocity distribution and the mixing time of the water body [38,39]. The residual bait and metabolic wastes produced by fish farming may also affect water quality. As the culture density increases, the mean velocity decreases and the turbulence characteristics are stronger. The resuspension of materials means that the self-purification capability is higher at high stocking densities [40,41]. It is recommended that experiments that include fish be conducted to explore the potential effects of their presence on the system.

This experiment was performed under the conditions of $30 \text{ l min}^{-1}$ aeration rate and an installation height at 10 cm from the bottom. Different test configurations would result in different hydrodynamic responses. Oca *et al.* [42–44] explored the effect of baffles, inlet characteristics, water depth, flow rate and other factors on the velocity distribution in different tank geometries. It is recommended that comparative tests on various design parameters of the system be conducted to determine the optimal design configurations that meet the economic requirements of the aquaculture industry.

The aeration scale remains unclear, making it necessary to conduct field measurements or to use a numerical simulation to explore the difference in aeration scale between the model and the prototype.

# 4. Conclusion

In our IPRS, the APFD was the only source of power. This feature differentiates the present system from previous systems using a specific flow pumping device. The APFD simultaneously increased the DO and promoted the circulation of the IPRS. The IPRS is in common use, and the flow conditions are important for optimization of the system. The flow field of the aquaculture raceway with APFD has been described in detail in this study. The main conclusions are as follows:

(1) At the fore-mid part of the raceway, both plug-flow and reflux were present. Flow was affected by turbulence. At the aforementioned midpoints, flow velocities and turbulence characteristics were higher than at the rear part, and values increased with height. At the rear part, the flow was uniform, with little change in any direction.
(2) The fluid exhibited obvious secondary flow, both in horizontal and some cross-sections. The secondary flow in horizontal sections was divided into two parts. At the fore-mid part, it was mainly induced by turbulence; in the rear part, it was caused by centripetal force.
(3) The velocities at the inlet and outlet of the raceway maintained a positive velocity value, indicating that the system could effectively self-circulate. The plug-flow capacity of the APFD was described. The reflux range reached approximately $x = 100$ cm, and the ratio between the area of plug-flow and the reflux was approximately $6:4$ at each height. The average plug-flow rate in the raceway was $994 \text{ cm}^3 \text{ s}^{-1}$ and the water retention time was about 3.8 min in our experimental model.

Although there were limitations to the present study, our findings advance the understanding of the hydrodynamic characteristics induced by APFD in IPRS and provide calibration and comparative data for future numerical simulation. The findings should enable optimization of IPRS design, thus improving this new mode of intense aquaculture.

Data accessibility. Our data are submitted to Royal Society Open Science as electronic supplementary material.

Authors' contributions. W.L. carried out in-laboratory measurement, analysis of experimental data and wrote the paper; X.C. participated in the design of the study and revision of the paper; J.X. contributed to the analysis of experimental data and generated the graphics; Z.W. carried out the statistical analyses of the data of water velocities and revised the manuscript; D.Y. designed the experiment and provided guidance. All authors gave final approval for publication.

Competing interests. We have no competing interests.

Funding. This work was supported by the National Natural Science Foundation of China (51579106); the China Modern Agro-industry Technology Research System (CARS-4617); the Special Fund for Economic Development of Guangdong province (SDYY-2018-07) and the Open Research Fund Program of State Key Laboratory of Hydraulics and Mountain River Engineering, Sichuan University (Skhl1815).

Acknowledgements. We thank Jiachun Hu for operational guidance and for providing experimental photos.

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
