## [Reviewer comments · Royal Society Open Science]

Review History

RSOS-182061.R0 (Original submission)

Review form: Reviewer 1

Is the manuscript scientifically sound in its present form?

Yes

Are the interpretations and conclusions justified by the results?

Yes

Is the language acceptable?

No

Is it clear how to access all supporting data?

No

Do you have any ethical concerns with this paper?

No

Have you any concerns about statistical analyses in this paper?

No

Recommendation?

Major revision is needed (please make suggestions in comments)

Comments to the Author(s)

This is a generally well-thought out paper that ties fundamental hydraulic understanding and experiments to a practical problem, achieving better conditions in an aquaculture pond. I make extensive comments and suggestions in the attached pdf (Appendix A) for the authors' consideration. In my opinion, the research is well justified, novel, and worthy of publication once language editing is complete, and the authors address these general questions/comments/concerns:

1) Depth as described in the paper is a bit confusing, since it is measured from the bottom, which is the opposite of its meaning. I suggest the authors check its use, and provide a bit more clarification so as to not confuse readers.

2) If possible, could each of the figures contain the legend on the figure/plot itself?

It would be easier for the reader.

3) In several cases, flow and velocity were interchanged, in most cases, I believe velocity is the appropriate term and was selected in the pdf. Please check and comment.

Review form: Reviewer 2 (Gargi Das)

Is the manuscript scientifically sound in its present form?

No

Are the interpretations and conclusions justified by the results?

No

Is the language acceptable?

No

Is it clear how to access all supporting data?

No

Do you have any ethical concerns with this paper?

No

Have you any concerns about statistical analyses in this paper?

No

Recommendation?

Major revision is needed (please make suggestions in comments)

Comments to the Author(s)

The study describes an interesting work on the influence of aeration plug flow device on the hydrodynamics of an in a pond raceway system. The work is experimental and an ADV has been used to obtain the mean & fluctuating velocities. The concerns are as follows.

1. The studies are specific to the system considered. It detects the primary & secondary flows but does not highlight the physics of flow. I am not sure how the results can be extended to other systems / in different geometry and/or different dimensions of the pond. In short I could not appreciate the novelty of the study and/or the generality and applicability of the results.
2. It is expected that aeration will produce secondary flow, but how is this information going to be useful in optimizing design of raceway ponds.
3. The ADV is also expected to disturb the existing flow. How does the author correlate the obtained flow characteristics with the actual flow characteristics expected in a pond.
4. There are several grammatical errors which makes understanding & appreciating results difficult. The paper has to be rewritten in correct English.
5. The hole dimension is not mentioned in experimental description
6. What does plug flow mean? This needs to be described as plug flow in multiphase flow denotes a completely different flow pattern.
7. How do the aeration orifices close by themselves? The mechanism is not clear from text.

Decision letter (RSOS-182061.R0)

15-Mar-2019

Dear Miss Cheng,

The editors assigned to your paper ("Hydrodynamics in an in-pond raceway system with aeration plug-flow device: experimental studies") have now received comments from reviewers. We would like you to revise your paper in accordance with the referee and Associate Editor suggestions which can be found below (not including confidential reports to the Editor). Please note this decision does not guarantee eventual acceptance.

Please submit a copy of your revised paper before 07-Apr-2019. Please note that the revision deadline will expire at 00.00am on this date. If we do not hear from you within this time then it will be assumed that the paper has been withdrawn. In exceptional circumstances, extensions may be possible if agreed with the Editorial Office in advance. We do not allow multiple rounds of revision so we urge you to make every effort to fully address all of the comments at this stage. If deemed necessary by the Editors, your manuscript will be sent back to one or more of the original reviewers for assessment. If the original reviewers are not available, we may invite new reviewers.

- Data accessibility

<http://datadryad.org/submit?journalID=RSOS&manu=RSOS-182061>

- Competing interests

- Authors' contributions

- Acknowledgements

- Funding statement

on behalf of Professor R. Kerry Rowe (Subject Editor)
openscience@royalsociety.org

Associate Editor's comments:

Two reviewers have commented on your paper, each recommending a number of modifications to the work to ensure it is publishable. Please note that the journal does not allow multiple rounds of revision, so please ensure that you follow the advice of the reviewers carefully, and incorporate the changes they recommend, as well as providing a full response in your response to referees. If you do not satisfy the reviewers that their concerns have been adequately addressed, we may not be able to consider your paper further.

Comments to Author:

Reviewers' Comments to Author:
Reviewer: 1

Comments to the Author(s)

This is a generally well-thought out paper that ties fundamental hydraulic understanding and experiments to a practical problem, achieving better conditions in an aquaculture pond. I make extensive comments and suggestions in the attached pdf for the authors' consideration. In my opinion, the research is well justified, novel, and worthy of publication once language editing is complete, and the authors address these general questions/comments/concerns:

- 1) Depth as described in the paper is a bit confusing, since it is measured from the bottom, which is the opposite of its meaning. I suggest the authors check its use, and provide a bit more clarification so as to not confuse readers.
- 2) If possible, could each of the figures contain the legend on the figure/plot itself? It would be easier for the reader.
- 3) In several cases, flow and velocity were interchanged, in most cases, I believe velocity is the appropriate term and was selected in the pdf. Please check and comment.

Reviewer: 2

Comments to the Author(s)

The study describes an interesting work on the influence of aeration plug flow device on the hydrodynamics of an in a pond raceway system. The work is experimental and an ADV has been used to obtain the mean & fluctuating velocities. The concerns are as follows.

1. The studies are specific to the system considered. It detects the primary & secondary flows but does not highlight the physics of flow. I am not sure how the results can be extended to other systems / in different geometry and/or different dimensions of the pond. In short I could not appreciate the novelty of the study and/or the generality and applicability of the results.
2. It is expected that aeration will produce secondary flow, but how is this information going to be useful in optimizing design of raceway ponds.

3. The ADV is also expected to disturb the existing flow. How does the author correlate the obtained flow characteristics with the actual flow characteristics expected in a pond.
4. There are several grammatical errors which makes understanding & appreciating results difficult. The paper has to be rewritten in correct English.
5. The hole dimension is not mentioned in experimental description
6. What does plug flow mean? This needs to be described as plug flow in multiphase flow denotes a completely different flow pattern.
7. How do the aeration orifices close by themselves? The mechanism is not clear from text.

Author's Response to Decision Letter for (RSOS-182061.R0)

See Appendix B.

RSOS-182061.R1 (Revision)

Review form: Reviewer 1

Is the manuscript scientifically sound in its present form?

Yes

Are the interpretations and conclusions justified by the results?

Yes

Is the language acceptable?

Yes

Is it clear how to access all supporting data?

Yes

Do you have any ethical concerns with this paper?

No

Have you any concerns about statistical analyses in this paper?

No

Recommendation?

Accept with minor revision (please list in comments)

Comments to the Author(s)

1. Lines 22-24, restate, sentence makes no sense. Plug-flow is not defined, is this the standard definition of plug-flow (note later use). I believe you intended for the two points to contrast strong vs. weak plug-flow, but this is not what is stated.
- 2, Line 62, what reaction mechanism?
3. Line 72, note plug-flow is in quotes. This indicates a special meaning; how does that differ from the standard meaning of plug-flow, a commonly used term? Please check all references to

the same term for consistency. If your intention was to use the standard meaning, then simply remove the quotes.

Decision letter (RSOS-182061.R1)

10-May-2019

Dear Miss Cheng:

On behalf of the Editors, I am pleased to inform you that your Manuscript RSOS-182061.R1 entitled "Hydrodynamics in an in-pond raceway system with aeration plug-flow device: experimental studies" has been accepted for publication in Royal Society Open Science subject to minor revision in accordance with the referee suggestions. Please find the referees' comments at the end of this email.

The reviewers and Subject Editor have recommended publication, but also suggest some minor revisions to your manuscript. Therefore, I invite you to respond to the comments and revise your manuscript.

- Ethics statement

- Data accessibility

If you wish to submit your supporting data or code to Dryad (<http://datadryad.org/>), or modify your current submission to dryad, please use the following link:
<http://datadryad.org/submit?journalID=RSOS&manu=RSOS-182061.R1>

- Competing interests

- Authors' contributions

- Acknowledgements

- Funding statement

Because the schedule for publication is very tight, it is a condition of publication that you submit the revised version of your manuscript before 19-May-2019. Please note that the revision deadline will expire at 00.00am on this date. If you do not think you will be able to meet this date please let me know immediately.

- 1) A text file of the manuscript (tex, txt, rtf, docx or doc), references, tables (including captions) and figure captions. Do not upload a PDF as your "Main Document".
- 2) A separate electronic file of each figure (EPS or print-quality PDF preferred (either format should be produced directly from original creation package), or original software format)
- 3) Included a 100 word media summary of your paper when requested at submission. Please ensure you have entered correct contact details (email, institution and telephone) in your user account
- 4) Included the raw data to support the claims made in your paper. You can either include your data as electronic supplementary material or upload to a repository and include the relevant doi within your manuscript

5) All supplementary materials accompanying an accepted article will be treated as in their final form. Note that the Royal Society will neither edit nor typeset supplementary material and it will be hosted as provided. Please ensure that the supplementary material includes the paper details where possible (authors, article title, journal name).

on behalf of Prof R. Kerry Rowe (Subject Editor)
openscience@royalsociety.org

Associate Editor Comments to Author:

The remaining comments from the reviewer are largely concerning the quality of the writing, which is a little disappointing to see, as this review also commented on the language in the first round of review: the authors should have responded better to this.

Nevertheless, as the reviewers seem to be largely satisfied that the paper is scientifically ready for acceptance, the Editors will be willing to accept the paper IF - and this is stressed - if the authors seek professional language advice: <https://royalsociety.org/journals/authors/language-polishing/>.

The authors will be required to provide evidence that such advice has been sought, in addition to the full point-by-point response to the reviewer's commentary.

Please ensure you seek the required advice.

Reviewer comments to Author:

Reviewer: 1

Comments to the Author(s)

1. Lines 22-24, restate, sentence makes no sense. Plug-flow is not defined, is this the standard definition of plug-flow (note later use). I believe you intended for the two points to contrast strong vs. weak plug-flow, but this is not what is stated.
- 2, Line 62, what reaction mechanism?
3. Line 72, note plug-flow is in quotes. This indicates a special meaning; how does that differ from the standard meaning of plug-flow, a commonly used term? Please check all references to the same term for consistency. If your intention was to use the standard meaning, then simply remove the quotes.

Author's Response to Decision Letter for (RSOS-182061.R1)

See Appendix C.

Decision letter (RSOS-182061.R2)

07-Jun-2019

Dear Miss Cheng,

I am pleased to inform you that your manuscript entitled "Hydrodynamics of an in-pond raceway system with an aeration plug-flow device for application in aquaculture: an experimental study" is now accepted for publication in Royal Society Open Science.

on behalf of Mr Andrew Dunn (Associate Editor) and R. Kerry Rowe (Subject Editor)
openscience@royalsociety.org

Associate Editor Comments to Author (Mr Andrew Dunn):
Associate Editor: 1
Comments to the Author:
(There are no comments.)

Reviewer comments to Author:

Appendix A**ROYAL SOCIETY
OPEN SCIENCE****Hydrodynamics in an in-pond raceway system with aeration
plug-flow device: experimental studies**

Journal:	Royal Society Open Science
Manuscript ID	RSOS-182061
Article Type:	Research
Date Submitted by the Author:	08-Dec-2018
Complete List of Authors:	Li, Wuhua; South China University of Technology Cheng, Xiangju; South China University of Technology, Xie, Jun; Chinese Academy of Fishery Sciences Pearl River Fisheries Research Institute Wang, Zhaoli; South China University of Technology Yu, Deguang; Chinese Academy of Fishery Sciences Pearl River Fisheries Research Institute
Subject:	ecology < BIOLOGY, environmental science < BIOLOGY, Limnology < EARTH SCIENCES
Keywords:	velocity, turbulence, aquaculture, recirculating flow, air micro-bubbles
Subject Category:	Engineering

Hydrodynamics in an in-pond raceway system with aeration plug-flow device: experimental studies

Wuhua Li¹, Xiangju Cheng^{1*}, Jun Xie², Zhaoli Wang¹, Deguang Yu²

¹ School of Civil Engineering and Transportation, South China University of Technology, Guangzhou 510640, China

² Pearl River Fisheries Research Institute, Chinese Academy of Fishery Science, Guangzhou 510380, China

Abstract: The in-pond raceway system (IPRS) is an effective intensive aquaculture mode coping with continuous consumption of water and land resources, which is widely used in many countries and regions. Water flow and dissolved oxygen (DO) are two important requirements for aquaculture. Several alterations made in the system design and the operational procedures to increase the water exchange and the DO concentration in the IPRS. The IPRS with aeration plug-flow device (APFD) is commonly used in China, where the APFD is the only stimulus for pushing water recirculation in an aquaculture pond. In order to further improve the design of the IPRS with APFD and to promote a novel intensive aquaculture mode, designers or farmers require comprehensive knowledge about the hydrodynamics in this system. The major experimental studies were performed and the three-dimensional velocities were measured by using an Acoustic Doppler Velocimeter. The patterns of velocity and the turbulent characteristics were analyzed and the plug-flow characteristics were analyzed. The results showed that the patterns of the velocity and the turbulence in the horizontal sections were divided into two parts. At one of parts there were plug-flow and the reflux and the value was higher and increased with depth from bottom, while at the other part the flow was mostly plug-flow and the distributions were uniform. Secondary flows were observed in both the horizontal and cross sections, which were beneficial to the diffusion of materials and the retention of DO. The system effectively self-circulated and the plug-flow capacity can be connected to a commercial scale to meet the general requirements

**Key words:** velocity; turbulence; aquaculture; recirculating flow; air micro-bubbles

1 Introduction

As China's economy ^{has} and aquaculture ^{is} developing rapidly, China has become

the world's largest producer and consumer of aquaculture products [1,2]. The

~~continuous consumption~~ of farm and water resources inevitably ~~accompanies this~~, so

34 ^{intensive} aquaculture ^{is} ~~plays~~ an important ~~role among the variable strategies~~ for

~~replacing the shortage~~ of resources. ~~The current traditional intensive aquaculture is~~

gradually exposing a series of problems, such as higher ammonia and nitrogen

concentration ^s [3], water waste, water pollution [4], and recurrent fish diseases [5].

The aquaculture water body ^{is} the ^{main} ^{habitat} ^{of} ^{fish} ^{and} ^{other} ^{organisms} ^{who} ^{live} as well as the decomposition

site of feces, bait, and the cultivation pool of plankton, so the function division of the

consumer, the producer, and the decomposer in aquaculture ponds remains unclear,

making the ~~management~~ difficult and ecological imbalance inevitable [6]. ^{mixed} ^{To cope with} ^{these} ^{challenges,}

an advanced aquaculture mode is required, where the ecological function zones remains

independent. ~~The fish production will improve and the water quality will be purified.~~

In 2005, Chappell ^{et} in Auburn University and a ^{local} farmer invented an in-pond

raceway system (IPRS) ~~cooperatively~~ [7], which is a simplified version of the

partitioned aquaculture system (PAS) that originated from Clemson University [8,9].

~~The basic principle of the IPRS~~ ^{consists of} is to ~~build~~ several aquaculture raceways ^{installed} in a pond.

The breeding area and the purification area become isolated, where the raceways

continue to provide "captive" to the feeding fish, and the purification area provides a

home for some filter-feeding fishes and grows ^{other} aquatic organisms [7-10]. In this mode,

it is easier to control fish excrement during the aquaculture period. By adding pumps

and waste collection equipment, ~~the~~ excreta and residual feed accumulates in the

downstream purification area for easier collection. ^{abstractment or removal.} The initial purpose of the IPRS ^{was} is

to improve the catching rate [10,11]. ^{IPRS is} ~~This mode~~ also suitable for the countries or

regions where high land use costs or environmental pressures exist [9,12]. The USA

has 12 large aquaculture fields using this ~~farming mode~~ [13]. In 2013, Zhou, who was

57 working in the United States Soybean Association (USSA), introduced ~~this mode~~ ^{IPRS} into
58 China, which was then ~~extended to~~ ^{implemented in} Jiangsu, Anhui, Shanghai, and other places (Chen,
2014). ~~The IPRS has been extended to more than 18 provinces and cities with an~~ ^{since}
aquaculture area of 200,000 m² in China. ^{footprint}

In IPRS, the water flow is one of the important environmental factors that affect
the behavior of fish. It stimulates the sensory of fish organs, produces the
corresponding reaction mechanism [14-17], and even generates behavioral variability
within a species ^{is generally due to} in response to ^{distributions} the velocity and turbulence [18]. ~~The~~ ^T turbulence plays
a critical role in the transport and dispersal of excreta, nutrients, and pollutants ~~in the~~
~~flow~~ [19,20]. Dissolved oxygen (DO) is an indispensable factor ^{to consider} in aquaculture ponds.

When DO concentration ^{is} in the water body ^{is} ~~less than~~ ^{droplet below} 5 mg/L, fish will exhibit lazy
behavior, ~~no~~ ^{do not respond to} ~~intaking~~ bait, and head ~~floating~~ ^{??}, possibly even suffocating, resulting in
significant economic losses [21-23]. In IPRS, one or more high-power water pumps

are used to pump water into the raceways. The water then flows out the raceways and
into the purification area, which circulates ^{the} water flow. ~~The water body in the~~ ^{does not}
~~raceways~~ ^{maintains} 24 hours of movement, forming an ~~artificial movement~~ ^{new path} ~~environment~~ ^{for the} fish. After several additional alterations were made in the system ^{does not make sense}

~~environment~~ for the fish. After several additional alterations were made in the system
design and the operational procedures, the water exchange and DO concentration ^S

increased in the IPRS. An IPRS (Fig. 1) with ^{an} ~~aeration~~ ^{new} plug-flow device (APFD) was

designed by some farmers in China, ~~where~~ ^{the} APFD ^{consists of} primarily includes a curved

baffle and a set of micro-bubble tubes. The APFD is fixed on the inlet of the raceway

and the micro-bubble tubes submerged ^{at specific} in a ~~certain~~ ^{depth} of ~~water~~, so that the air

through the external compressor is pumped into the micro-bubble tubes, resulting in

micro-bubbles floating into water. The floating micro-bubbles are blocked by the top

of the curved baffle and forced to change their original floating path, which causes the

water current to flow from the bottom to the top, from the raceway inlet to the outlet,

and finally to the purification area. This forms a recirculating flow in the IPRS, and

mass transfer between the interface of the micro-bubbles and the water ~~takes place~~, so ^{of oxygen occurs}

^{resulting in}

~~that~~ the DO concentration increases in the water. The IPRS with APFD enables water
 recirculation in an aquaculture pond and supplies DO to the water ~~at the same time,~~
 making it increasingly popular in ~~many~~ Chinese aquaculture ponds. However there is
 a lack of sufficient research exploring not only the effects that APFD has on the
 hydrodynamic characteristics in raceways, but also the ~~plug-flow capacity~~ of the
 APFD in IPRS. Farmers often use IPRS with APFD blindly. There is confusion about
 how the hydrodynamics are in raceways under a ~~certain~~ air flow rate and how to set
 up or improve this device more efficiently. At some sites, the water recirculation
 velocities or turbulence exceed the preferable conditions for fishes, which could
 compromise fish production. In order to understand the response of the
 hydrodynamics to the APFD in raceways, we designed an indoor physical model of
 the IPRS with APFD, regulated an aeration flow rate, performed a series of velocity
 measurements in the raceway, analyzed the distribution of velocity and turbulence,
 and ~~at last~~ calculated the plug-flow capacity of the APFD. This study could provide a
 reference for future optimization design of the IPRS and promote the new intensive
 aquaculture model.

Handwritten notes in red:
 and applies DO to
 simultaneous
 new
 improvements in
 that
 The full potential of the APFD innovation cannot be realized without better understanding of underlying hydraulic characteristics as the basis
 and course economic losses
 specific
 conditions of local
 APFD
 and
 characteristics
 and

Fig. 1 Diagram of the IPRS with APFD.

2 Experimental setup and methods

2.1 The aeration plug-flow device (APFD)

A detailed structure of the APFD is shown in Fig. 2. There is the framework (1),
 an aeration device (2), and an arc-shaped baffle (3). The aeration device included a
 gas pipes group (4) and a micro-bubble generating tubes group (5). One pipe in the
 gas pipes group was composed of an air inlet ~~hole~~ ^{an} (6) and a gas transmission pipe (7).
 The micro-bubble generating tubes group included tee pipes or adapters (8), five
 micro-bubble generation tubes (9), and gas nozzles. The gas nozzles were connected
 ~~together~~ ^{set to be} to the tee pipes or adapters through the internal thread.

The width of the APFD was ~~exactly~~ ^{set to be} the same as the width of the raceway. The
 APFD was fixed at the inlet and near the bottom of the raceway, air pumped by an
 external air compressor passed through the air inlet hole (6), the gas transmission pipe
 (7), the tee pipes or adapters (8), the gas nozzles, ~~and~~ ^{ing} eventually reached the
 micro-porous aeration tubes (9). This generated a large number of tiny bubbles
 floating from the bottom.

**Fig. 2.** Detailed structure of the aeration plug-flow device.

(1-framework, 2-aeration device, 3-arc-shaped baffle plate, 4-gas pipe group, 5- micro-bubble
 generating tubes group, 6-inlet hole, 7-gas transmission pipe, 8-tee pipe or adapter, and 9-
 micro-porous aeration tubes).

~~Fig. 3~~ shows the actual appearance of the micro-porous aeration tubes. The
 aeration tubes were made of a nano-polymer material with stable physical and
 chemical properties, with outer and inner diameters of 15 mm and 10 mm, respectively. The surface
 of the aeration tube had a number of tiny holes, with a density between 700 to 1200
 holes per meter length. Once the air was pumped into the aeration tube, the increased
 air pressure opened the aeration orifices; otherwise ~~they~~ ^{were} shut off by themselves. The
 ~~designed orifices could~~ effectively avoid clogging by sediments or dusts. The air
 bubbles produced from this aeration tube were small enough to enhance the total
 contact area between the air bubbles and the water.

is shown in Fig. 3

respectively.

automatically
this
feature

Fig. 3. Actual appearance of the micro-pore aeration tube (taken by Hu).

2.2 The indoor in-pond raceway system (IPRS)

~~Fig. 1~~ shows that our indoor IPRS model consisted of three APFDs, three
 aquaculture raceways, and a purification area. The four sides and the bottom of the
 model were reinforced by concrete. The size was 370cm×310cm×75cm (length ×
 width × height) and the concrete wall thickness was 10cm and higher than the water
 level, which ensured the water ~~did~~ ^{is} not overflow. There was a walkway to facilitate the
 experimental operation in between the two raceways.

~~The equipment and size~~ ^{As a cobite} of the raceway ^{is} are shown in Fig. 4. The size of each
 raceway was 220cm×30cm×40cm (length × width × height). Each raceway was made

as shown in Fig. 1

with

could

was installed

of plexi-glass, where the entrance and exit of the raceway both connected with the
purification area (Fig. 1). The APFD is ~~the only impetus~~ for the circulatory system
and was installed at the ~~front~~ front of the raceway and fixed 10cm from the bottom of
~~the raceway~~.

After turning on the air compressor and adjusting the pressure and air flow rate,
the micro-porous aeration tube continued to produce a large number of floating
micro-bubbles from the bottom. When the bubbles were subjected to the obstruction
of the upper arc-shape baffle, they were forced to change the original floating path,
flowing out from the front of the curved baffle, while influencing other floating
bubbles to change their path (shown in Fig. 4b). The APFD caused the air bubbles to
flow from the bottom to the surface, from the inlet to the outlet of the raceway, which
drove the water to move through the aquaculture raceway, passing through the
purification area, and returning to the inlet of the raceway. As shown in the red arrows
of Fig. 1, the water of the entire system circulated clockwise.

Fig. 4. (a) Equipment and sizes of aquaculture unit raceway (A-A profile of Fig. 1).
(b) Photo of the bubble movement (taken by Hu).

2.3 Flow velocity measurement

This research focused on the ~~response~~ ^{effect} of the APFD on the hydrodynamics in ~~the~~ ^{the}
 IPRS, so ~~the~~ ^{the} fish ~~were~~ ^{arent?} not bred in the raceway to avoid ~~influencing~~ ^{complication} the
 hydrodynamics. ~~The~~ ^{Since the} three parallel raceways had similar flow fields, so just one of the
 ~~raceways~~ was chosen for measurement.

Measurements of the 3D velocities were conducted via Nortek Acoustic Doppler
 Velocimeter (ADV) and we designed a custom displacement device. The
 displacement device comprised of a base, a sliding rod, a lifting rod, and a fixing part.
 (Fig. 5). The ADV was mounted on the ~~fixing~~ ^{ed} part and could slide freely along the
 longitudinal, transversal, and vertical directions of ~~the~~ ^{the} raceway to measure any point in
 the water.

**Fig. 5.** Diagram of the displacement device.

The direction of the ADV probe affected the Reynolds ~~stress~~ ^{shear} and the secondary
 flow, so to minimize the error from the probe, the positive direction of the probe was ~~set~~ ^{set}
 in the positive direction of ~~the~~ ^{with} ~~x-axis~~ ^{as shown in Fig. 6.}. The ~~x~~ direction was parallel along the raceway;
 the ~~z~~ direction was vertical along the ADV rod; the ~~y~~ direction was determined by the
 right hand law and was 90° angle with x direction (Fig. 6). ^{data}

Referring to the division of the water in ~~open-channel shear flow~~ ^{use flow an} into three
 regions studied by Nezu and Nakagawa [24,25] and considering our experimental
 conditions, sets of measurements in depths from the bottom $z=5$ cm, 10 cm, 15 cm, 20
 184 cm, 25 cm, and 30 cm were collected, with relative depths of 0.15, 0.29, 0.44, 0.59,
 0.74, and 0.88, as shown in Table 1. ^{corresponding}

For each set of measurements, ADV data was sampled at the frequency of 25 Hz
 within a regular grid consisting of 12 laterals of 6 points spaced 5 cm apart, producing
 a grid of 72 points (Fig. 6), and the signal-to-noise ratio (SNR) was maintained at 15
 or above. The ADV sampling was performed for a minimum of 60 s at each point,
 with an average sampling time of 100 s.

 **Fig. 6.** Definition sketch of the coordinate system and arrangement of measured points.

**Table 1** Measurement sets and reference division.

Depth from bottom z (cm)	Relative depth z/h (water depth $h=34$ cm)	Nezu and Nakagawa's division of the water in open-channel shear flow	
0.15	$z/h=0-0.15$	Wall region
0.29
0.44	$z/h=0.15-0.6$	Intermediate region
0.59
0.74
0.88	$z/h=0.6-1.0$	Free surface region

2.4 Data filtering and processing

To ensure the authenticity of the measured data, ~~the data where the correlation~~
 coefficient was less than 80% ~~were eliminated~~. About 2000 ~~valid data~~ at each point
 ~~were taken to calculate~~.

me eliminated from consideration instances
measurements
(this still left approximately)

198 Instantaneous 3D flow velocity components were acquired at each measurement
 point using the ADV. ~~The measurement data~~ *you make* in the longitudinal, the lateral, and the
 vertical directions ~~were~~ *u* denoted as u_i , v_i , and w_i . The instantaneous velocity ~~was~~ *no can be*
 divided into the mean velocity and the fluctuating component, as shown in the Eq. (1).

$$u_i = u + u_i'$$
 (1a)

$$u_i' = u_i - u$$
 (1b)

where u_i is the instantaneous velocity, u is the mean velocity, and u_i' is the
 fluctuating velocity. The mean velocity u ~~was~~ *can be* obtained by Eq. (2).

$$u = \frac{1}{N} \sum_i^N u_i$$
 (2)

where N is the number of valid data. The ~~turbulence~~ *can be* strength of the longitudinal
 velocity ~~was~~ *can be* defined as:

$$u' = \sqrt{u_i'^2} = \sqrt{\frac{1}{N} \sum_i^N (u_i')^2} = \sqrt{\frac{1}{N} \sum_i^N (u_i - u)^2}$$
 (3)

Similar definitions ~~applied~~ *can be* to the lateral and vertical velocities v_i and w_i .

The turbulent shear stress (i.e., Reynolds shear stress) ~~were~~ *are* calculated as:

$$\tau_{uv} = |u'v'|, \tau_{uw} = |u'w'|, \tau_{vw} = |v'w'|$$
 (4)

where $u'v'$, $u'w'$, and $v'w'$ are the covariance of instantaneous fluctuating
 velocity.

~~And the~~ *is* turbulent kinetic energy k ~~was~~ *per unit mass?*:

$$k = \frac{1}{2}(u'^2 + v'^2 + w'^2)$$

For all equations and definitions include units!

**3 Results and discussion**

To understand the ~~distributions~~ *velocity* of the velocity and the turbulence in the raceway,
 the patterns of mean velocity, the Reynolds shear stress, and ~~the~~ *the* turbulent kinetic

energy were derived from the ADV measured data. The APFD induced ~~the~~ water flow ~~to move~~ from the bottom to the surface, from the inlet to the outlet of the raceway, so we were primarily concerned ~~about~~ ^{with} the velocity patterns in the horizontal sections and the cross-sections.

3.1 Velocity patterns in horizontal sections

The schematic diagram of various parts in ~~the~~ ^{the} horizontal section is ~~shown~~ ^{are} in Fig. 7(i), where the sections of $x=-40$ and $x=180$ were the inlet and the outlet, ~~for the~~ ^{are} convenience of the following description, $x=0-120$ was defined as the fore-mid part and $x=120-185$ was defined as the rear-part. ^{which is it?}

As shown in Fig. 7(ii), ~~the~~ ^{the} flow velocity patterns in horizontal sections were plotted where (a)-(f) represented the distributions of velocity at different depths $z=5$, 10 cm, 15 cm, 25 cm, and 30 cm. The velocities at the baffle were corrected to zero. The color indicated the velocity value of $u+v$, the blue vector arrows indicated ~~the~~ ^{the} direction of $u+v$, the gray dotted lines denoted the isoline of $u+v=0$, and the shaded part denoted the APFD's position. ^{note present tense when discussing figures}

Fig. 7. (i) Schematic diagram of various parts in the horizontal section.

(ii) Velocity patterns in the horizontal sections.

Color represents the horizontal patterns of the velocity. Vectors describe the direction of the horizontal velocity (u,v) . The gray dotted lines denote the isoline of $u + v = 0$.

Fig. 7 (ii) shows that the flow in the raceway was divided into plug-flow ($u+v > 0$) and the reflux ($u+v < 0$), where the gray dotted lines were the boundary lines. The range of the reflux and the area proportions between the plug-flow and the reflux are listed in Table 2. The range of reflux extended to about $x=100$, and the area ratio was approximately 6:4. In the plug-flow area, the water was pushed through the raceway for self-circulation, while the reflux contributed to retaining the DO in the aquaculture unit for fish consumption.

Table 2. The range of reflux and the ratio between plug-flow and reflux

Depth from bottom z (cm)	Range of reflux (cm)	Plug-flow proportion (%)	Reflux proportion (%)
($z/h=0.15$)	0-91	0.64	0.36
($z/h=0.29$)	0-77	0.68	0.32
($z/h=0.44$)	0-99	0.62	0.38
($z/h=0.59$)	0-114	0.57	0.43
($z/h=0.74$)	0-120	0.58	0.42
($z/h=0.88$)	0-108	0.65	0.35
Average	0-102	0.62	0.38

At each depth, the patterns at the fore-mid part ($x=0-120\text{cm}$) and the rear part ($x=120-180\text{cm}$) were distinct (Fig. 7). The plug-flow and the reflux interacted at the fore-mid part. The velocity was generally higher than the rear part. There were two cores of high velocities distributed at the fore-mid part, which were encircled by the diminishing velocities and separated on two sides of the raceway. The two cores formed by the different effects; the red core was formed by the plug-flow and the blue one was formed by the reflux. At the rear part, the velocity was relatively uniform, mostly showing the plug-flow effect, and the strength did not change much with depth. The velocities at the fore-mid part varied obviously with depth and the changes were mainly reflected in the movement of the high velocity cores, the magnitude of velocity, and the reflux range.

At the depth near bottom ($z=5$), the reflux zone was near the APFD, while the plug-flow zone was slightly behind. The plug-flow effect generally began at $x=10$. When approaching the water surface, i.e., at $z=25$, another plug-flow zone appeared at

Shear-O definition
in most
detail
some where.

the position of APFD. Until $z=30$ ^{cm} the two plug-flow zones developed and ~~connected~~ ^{merged}
 ~~together~~, showing that the pattern that the plug-flow zone was near the APFD while
 reflux zone was ~~the latter~~ ^{far from it.}. From the bottom to the water surface, ^{within} the velocity of the
 plug-flow and the reflux at the fore-mid part gradually increased, but the enlargement
 of reflux was less than ~~the~~ ^{regions that of} plug-flow. The reflux range at $z=10$ was the shortest
 among the measured depths (Table 2), even shorter than the range at $z=5$, which was
 below the installation height of the APFD. Above the installation height, the reflux
 range gradually increased at depths of $z=15-25$ and reduced to about 100 cm at $z=30$.
 At depths of $z=10$ and 30, the area ratio between the plug-flow and the reflux was
 ~~closed~~ ^{approximately} to 7:3.

The blue vectors represented the patterns of the secondary flows that occurred at
 each depth (Fig. 7). The secondary flow patterns was divided into the two parts and
 their rotation ^{at} directions were opposite, i.e., at the fore-mid part, the water flow rotated
 counterclockwise and at the rear part it rotated clockwise. The vectors showed that as
 the depth ^{increased} from the bottom to the surface, the secondary flow cell at the fore-mid part
 moved backward and the curvature of the vector at the rear part increased.

The velocities at the inlet and outlet of the raceway showed that most of the $u+v$
 values were more than zero, indicating that under the influence of APFD, the water
 was effectively pushed through the raceway, flowing into the purification zone, and
 returning to the aquaculture unit for self-circulation.

3.2 Velocity patterns in cross-sections

The velocity distributions in ~~the different~~ ^{each} cross-sections are shown in Fig. 8. The
 color showed the velocity value of $v+w$, the blue vector arrows indicated the direction
 of $v+w$. The gray dotted lines were boundaries between the plug-flow and the reflux
 which were consistent with the horizontal velocity patterns above. In reference to Fig.
 7(ii), the plug-flow and the reflux zone were distinguished and separated on two sides
 of the dotted line.

At the inlet and the rear part, i.e., ~~the~~ C-(40) and ~~the~~ C-140~185, the flow was
 mostly pushed and the velocities were uniform. At the fore-mid part (C-5~120), there
 were a plug-flow zone and a reflux zone. Velocities varied greatly with the
 cross-sections and were mostly larger at the plug-flow side. In the two cross-sections
 immediately downstream of the APFD, i.e., C-5 ~~and C-20~~, there were high velocity cores
 that were primarily distributed at the plug-flow zone. The low velocity cores were
 located in the reflux zone. The plug-flow area was smaller than the reflux area in
 these two sections. At the following sections, i.e., the C-40~120, the plug-flow area
 gradually expanded. There were low velocity cores distributed in the reflux zone and
 the velocities of the entire section tended to be uniform.

The vectors showed that there were secondary flows at the center and the flow
 near side walls of the raceway moved from the bottom to the surface (i.e., the large
 blue arrows in Fig. 8). The secondary flow at the center rotated counterclockwise and
 its cell approximately coincided with the low velocity core in the reflux zone (as
 shown by the yellow circle in Fig. 8).

Fig. 8. Velocity patterns in the cross-sections.

Color represents the cross-sectional velocity patterns. Vectors indicate the direction of the cross-sectional velocity (v, z). The gray dotted lines denote the isoline of $u + v = 0$.

3.3 Turbulent characteristics

In the self-aeration models with high water heads and flow rates, air bubbles play an important role in drag reduction and accelerate the dissipation rate of turbulent kinetic energy [26]. This is also the mechanism of erosion reduction and energy dissipation in some hydraulic structures. In this study, the movements of air bubbles

drove the water in the system to circulate. The existence of air bubbles should be positively correlated with ~~the water~~ turbulence.

In Fig. 9, the lines with scattered points represented the Reynolds shear stress in each direction, and the color indicated the magnitude of the turbulent kinetic energy. The sub-graphs (a)-(f) represented the distributions of turbulence characteristics at different depths $z=5$ cm, 10 cm, 15 cm, 25 cm, and 30 cm.

Fig. 9. Turbulence characteristics in the horizontal sections.

Color represents the horizontal patterns of the turbulent kinetic energy. Lines indicate the Reynolds shear force in the three directions.

At each depth, the turbulence characteristics were not evident at the inlet ($x=-40$) and the rear part ($x=140-185$), while appeared intensively at the fore-mid part ($x=0-120$) of the raceway. Combined with Fig. 7(ii), it shows that both the Reynolds shear stress (τ) and the turbulent kinetic energy (k) in the plug-flow zone were higher than in the reflux zone. The values of these turbulence characteristics gradually increased close to the water surface.

The patterns of the Reynolds shear stress (τ) in the three directions were similar. Besides the maximum value in the plug-flow zone, there was a smaller peak in the reflux zone. The Reynolds shear stress in each section were compared to show that the value of $|u'w'|$ was larger than that in the other two directions, while the values of $|u'v'|$ and $|v'w'|$ varied little, indicating that the turbulence in the longitudinal and vertical directions was relatively large and the strength did not differ much.

The turbulent kinetic energy (k) showed a single trend with water depth. There was only one core of high value that was located in the plug-flow zone. Its value decreased outward. Close to the free surface, the maximum of k increased and the core tended to move to the APFD. This movement trend was the same as the

to a maximum

movement of the high velocity core of plug-flow with ~~water~~ depth (shown in Fig. 7),
 showing that the velocity at the fore-mid part had ~~a good correlation~~ with the
 turbulence. *was correlated*

The experimental fluid was a type of bubble-induced multiphase flow in the
 circulating system. The movements of the bubbles induced ~~the flow~~ turbulence, and
 ~~by~~ centripetal force was generated at the turn of the circulation system. ~~There were~~
 ~~evident~~ secondary flows in the different sections. *were evident.* According to the mechanism, the
 ~~secondary flows were divided into two categories~~ by Prandtl [27]. *secondary flows can be* The secondary flow *divided into*
 caused by the centripetal force was termed the first kind which is common in river *two parts.*
 bends. However secondary flows can be observed in some straight channels because
 of ~~the~~ turbulence anisotropy. This kind of secondary flow was termed the second kind
 [28]. *for secondary flows are 2 kind* Its mechanism was first deduced by Einstein and Li [29] on the basis of RANS
 equations and many ~~current~~ numerical simulations tried to reproduce ~~this kind of~~ *attempted* ~~its~~
 ~~secondary~~ [28,30]. *shown in* Fig. 9 and Fig. 7 show that the flow turbulence was more intense
 at the fore-mid part and the secondary flow of the Prandtl's second kind ~~was~~ *is* obvious.
 The turbulence at the rear part greatly weakened and the centripetal force had a
 greater contribution to the water flow, primarily generating ~~the~~ secondary flows of
 Prandtl's first kind.

The patterns in depths of $z=5$ and $z=10$ ~~are~~ *are* worthy of attention. The depth of
 $z=5$ was located below the installation height of the APFD and the $z=10$ depth was the
 APFD installation height where the air bubbles generated and floated ~~there~~, thus these
 two depths were little affected by bubbles. Yet the patterns similar to the upper depths
 were observed in these two depths. This was because that there were secondary flows
 in ~~the~~ cross-sections (Fig. 8) so the turbulence was transmitted in the vertical direction
 and obvious turbulence remained even in the wall region (in reference to the division
 of Table 1).

3.4 Plug-flow characteristics

Fig. 8 shows that the velocities at the last three cross-sections ($x=160\sim 185$) were mostly positive and there was little reflux. To measure its cycle efficiency, the average flow rate of the three sections was used to represent the overall flow rate of the raceway. The effects of the lateral and vertical velocities were ignored (v and w), so the plug-flow rate was calculated using the streamwise velocity (u) according to Eq. (6).

$$Q_{160} = \iint u(160,y,z)dydz \quad (6)$$

Where Q_{160} is the flow rate at cross-section $x=160$, $u(160,y,z)$ is the streamwise velocity at each point of measurement point at $x=160$. The flow rates at cross-sections $x=175$ and $x=185$ i.e., Q_{175} and Q_{185} , were calculated the same way.

The plug-flow rates of the last three sections are seen in Fig. 10. The computing results of the flow rate at each cross-section are shown in Table 3. The calculated average flow rate was $994 \text{ cm}^3/\text{s}$ and the average streamwise velocity was $U = 0.97 \text{ cm/s}$.

The specific flow pattern in the raceway was neglected, so the cycle efficiency was represented by the average flow rate. The water retention time was calculated by Eq. (7),

$$t = V/Q \quad (7)$$

where V indicates the volume of water in raceway. The water retention time was calculated, $t \approx 226 \text{ s} \approx 3.8 \text{ min}$. There was reflux in the raceway, so the retention time was longer, and the calculated value was underestimated. This also meant that oxygen remained in the raceway for a long time, rather than being simply transferred downstream by the flow, which increased the efficiency of aeration.

Fig. 10. The integration of the last three cross-sections.

Table 3 Integral flow rate of the last three cross-sections.

Cross-section (cm)	Integral flow rate of each section (cm ³ /s)	Average value
x=160	902.8	Q=994cm ³ /s
x=175	994.1	
x=185	1087.0	

The IPRS is applied to commercial scales in the United States and China. The general engineering size in China is 20~25m length, 4~5m wide and 1.5~2m water depth. The longitudinal and lateral linear scales between the prototype and the model were $\lambda_L = 5/0.3 = 16.7$, and the vertical linear scale was $\lambda_V = 2/0.34 = 5.88$. To extend the experimental results into the field, the scale of the average flow rate was $\lambda_Q = \lambda_L \lambda_V^{1.5} = 238$, and the scale of the average velocity was $\lambda_v = \lambda_V^{0.5} = 2.4$. The average flow rate in the prototype reached $Q_P = 0.237\text{m}^3/\text{s} = 853\text{m}^3/\text{h}$ and the average velocity was about $U_P = 0.024\text{m}/\text{s}$. The length of each raceway primarily influenced the breeding capacity and density of the system, which had less influence on the flow rate and the exchange time, so it was rarely considered in the design. The longitudinal scale was ignored and we tried to convert the experimental model into

59

the commercial application of Brown et al. [12]. Each raceway size in Brown's
 experiment was 7.71m×4.88m×1.22m (length × width × depth). The lateral linear
 scale between the two models was $\lambda_L = 4.88/0.3 = 16.3$, and the vertical linear scale
 was $\lambda_V = 1.22/0.34 = 3.6$. The scale of flow rate was $\lambda_Q = \lambda_L \lambda_V^{1.5} = 111$. The
 conversion flow rate was $0.11\text{m}^3/\text{s}$, which was close to Brown's experimental result
 of $0.15\text{m}^3/\text{s}$. It shows that the APFD in the present experiment can effectively
 promote the water flow in a commercial scale.

~~The hydrodynamic characteristics in an IPRS with APFD are clearly understood.~~

However, although there were some deficiencies in this study.

The experiment was performed without fish, although fish exist in nature and

will affect the hydrodynamic and the water quality in the raceway. These conditions

will also affect the physiological function and behavior of the fish. The movement of

the fish increases the turbulence, which affects the hydrodynamic conditions, such as

the drag coefficient, the average velocity distribution, the turbulence characteristics,

and the retention time of the water body [31,32]. The residual baits and metabolic

wastes produced by fish farming could affect the water quality. As the culture density

increases, the mean velocity decreases, and the turbulence characteristics are higher.

The resuspension of the materials means that the self-purification ability is higher at

high stocking densities [33,34]. It is ~~necessary to conduct~~ experiments that include

~~fish breeding conditions~~ to explore the effect that fish have on the system.

This experiment was performed under the conditions of aeration rate at 30L/min

and the installation height at 10m from the bottom. Different test configurations

would result in different hydrodynamic responses. Oca et al. [35-37] ~~has~~ explored

the effect that baffle, inlet characteristics, water depth, flow rate, and other factors

have on the distribution of flow in different tank geometries. It is ~~necessary to conduct~~

comparative tests on various design parameters of the system to determine the

optimal design and configurations to meet the economic requirement ~~for~~ of the

aquaculture industry.

The IPRS is an abnormal model in laboratory. The aeration scale remains
unclear, making it necessary to conduct field measurements or to use a numerical
simulation method to explore the aeration scale between the model and the prototype.

**4 Conclusion**

In our IPRS, the APFD was the only source of power. This differs from the
previous system using the specific flow pumping device. The APFD simultaneously
increased the DO and promoted the circulation of the IPRS. ~~The~~ IPRS is commonly
used and the flow conditions are important to operate and optimize the system. The
flow field of the aquaculture raceway with APFD was described in detail in this study.

The main conclusions are as follows:

(1) At the fore-mid part of the raceway there were the plug-flow and the reflux.
~~The~~ flow was affected by the turbulence. ~~Both~~ ^{At reference midpoints,} flow velocities and the turbulent
characteristics were higher than the rear part and ~~the~~ values increased with depth. At
the rear part, ~~the~~ flow was uniform and changed little in all ~~directions~~ ^{any}.

(2) The fluid had obvious secondary flow in both the horizontal and ^{in some} cross
sections. The secondary flow in horizontal sections was divided into two parts. The
secondary flow at the fore-mid part was mainly induced by the turbulence and the rear
part was caused by the centripetal force.

(3) The velocities at the inlet and the outlet of the raceway maintained a positive
velocity value, indicating that the system could effectively self-circulate. The
plug-flow capacity of the APFD was described. The reflux range reached at about
$x=100$ and the ratio between the area of plug-flow and the reflux was about 6:4 at
each depth. The average plug-flow rate in the raceway was $994 \text{ cm}^3/\text{s}$ and the water
retention time was about 3.3 min in our experimental model.

Although there were some shortcomings, this study aided ⁱⁿ understanding ^{the} the
hydrodynamic characteristics induced by APFD in IPRS, ^{and} as well as providing ^{of} ~~as~~
calibration and comparative data for future numerical simulation. This could optimize
~~the~~ design of IPRS and improve ^{this} the new mode of ~~the~~ intense aquaculture ~~system~~.

**Ethics**

Our research were not required to complete an ethical assessment prior to
conducting study.

**Data accessibility**

Our data are submitted to Royal Society Open Science as electronic
supplementary material (ESM).

**Competing interests**

We have no competing interests.

**Funding**

This study was supported by the National Natural Science Foundation of China
(51579106), China Modern Agro-industry Technology Research System
(CARS-4617) and Provincial Special Project for Promoting Economic Development
in Modern Fisheries Development in 2018 (YY-201807).

**Acknowledgments**

We thank Jiachun Hu for operation guides and providing experimental photos.

**Authors' contributions**

Wuhua Li made the contributions on in-lab measurement, analysis of experimental
data and writing of the paper; Xiangju Cheng participated in the design of the study and
revising the paper; Jun Xie made the contributions on analysis of experimental data and
doing graphics; Zhaoli Wang and Deguang Yu made the contributions on providing
convenient laboratory and instruments for this experimental research. All authors gave
final approval for publication.

[revised manuscript text omitted]

630 <http://dx.doi.org/10.1016/j.aquaeng.2013.11.001>

Appendix B

Response to referees

Dear Dr. Editor,

Thank you so much for your E-mail letter dated March 15, 2019, regarding our manuscript entitled “*Hydrodynamics in an in-pond raceway system with aeration plug-flow device: experimental studies*” (Manuscript ID **RSOS-182061**). We deeply appreciate the referees’ scholarly review. We believe that the revisions have improved the quality of the manuscript substantially. Enclosed please find our response to the referee’s comments, and the revised manuscript, for consideration of publication in Royal Society Open Science.

Thank you.

Sincerely,

Xiangju Cheng.

E-mail: chengxiangju@scut.edu.cn

Response to reviewers’ comments

We thank the reviewers for their constructive comments on our manuscript.

Answers to Reviewer #1

- 1) As the depth described in the paper is a bit confusing, we change the description and use height instead.
- 2) We add a legend to each figure/plot under considering your suggestions.
- 3) We sincerely thanks for your careful review. We have modified the language and usage in the paper according to the pdf file you attached.

Answers to Reviewer #2

- 1) The main difference between the design in this manuscript and the other general IPRS is that the APFD pushes the water and provides DO. And similar designs have been widely used in China. However, in practice, as lacking of the understanding of the hydraulic response, the DO and water flow conditions in the water body cannot be properly adjusted. Because field trials are costly and difficult to find the impact of APFD on water flow, we referred to existing designs for indoor model testing to measure detailed and sufficient hydrodynamic data. Rectangular raceway saves 20% space than circular tank [1][2] and is easy to construct and reform, making it the most popular type in China. The experimental results can be converted to the commercial scale according to the similarity principle, and

will provide test data for further numerical simulation. Corresponding description can be found in Introduce section (*Line 87-92, 72-74, 101-103*)

- [1] Ronald M (2013) Recirculating aquaculture tank production systems a review of current design practice. SRAC Publication No. 453
- [2] Jacob B (2015) A Guide to Recirculation Aquaculture. FAO and EUROFISH
- 2) The secondary flow influences the suspended sediment concentration [3], affects the near bed pattern to some extent [4], and promotes the transport processes [5]. On the one hand, the secondary flow in cross-sections as well as the turbulence we observed in our experiments can prevent residual bait and feces from being deposited in the raceway. On the other hand, the secondary flow promotes the mixing of materials and energy. The use of secondary flow in optimizing design can make the materials diffuse and mix more fully, being utilized by fish. We added corresponding description at *Page 21, Line 373-377* in Results and discussion section.
- [3] ZhiQian W. and NianSheng C. (2008) Influence of secondary flow on distribution of suspended sediment concentration, *Journal of Hydraulic Research*, 46:4, 548-552
- [4] McLelland SJ, Ashworth PJ, Best JL, Livesey JR (1999) Turbulence and Secondary Flow over Sediment Stripes in Weakly Bimodal Bed Material. *Journal of Hydraulic Engineering* 125:463–473
- [5] Peterson EL (1999) Benthic shear stress and sediment condition. *Aquacultural Engineering* 21:85–111
- 3) The ADV we used in the experiment is Nortex Vectrino. It uses the Doppler effect to measure current velocity. Sound does not reflect from water, but from particles suspended in the water. As shown in **Fig. 1**, the Vectrino acoustic sensor consists of a center transducer and four receivers. The Vectrino transmits short pairs of sound pulses from the center transducer, listens to their echoes, and, ultimately, measures the change in pitch or frequency of the returned sound [6]. **Fig. 2** shows the beam crossing at approximately 50mm from the transducer, so the sampling point is far away from the sensor, thus avoiding the instrument's own interference with the water flow.

The Vectrino velocimeter operating principle: A pulse is transmitted from the centre transducer, and the Doppler shift introduced by the reflections from particles suspended in the water, is picked up by the four receivers.

Fig. 1 The Vectrino velocimeter operating principle [6] .

Fig. 2 The schematic of the sample point [6].

[6] Vectrino velocimeter user guide 2009

- 4) This article has been revised by a native speaker, please see the attached pdf. file.
- 5) The surface of the aeration tube has a number of tiny holes, with a density of 700 to 1200 holes per meter length. The average diameter of these holes is 50 μm . We appreciate your comments and added the information to the article. (*Page 6, Line 126* in Experimental setup and methods section.)
- 6) We refer to the Van's definition [7]. In a rectangular aquaculture raceway, the water flows from one end to the other end. This typical flow pattern is called “plug-flow”. Different from the plug flow in multiphase flow, its main feature is that the water flow is pushed and circulates under the pump or aerator. This definition is also cited in many publications on recirculating aquaculture system, such as the publications [7]-[11] listed below. Considering that the definition is a bit confusing, we added a description to this in the article (*Page 3, Line 71-72* in Introduce section).

[7] Van Wyk, P. (1999). Principles of recirculating system design. Pages 59–98 in P. Van Wyk, M. Davis-Hodgkins, R. Laramore, K. L. Main, J. Mountain, and J. Scarpa,

editors. Farming marine shrimp in recirculating freshwater systems. Florida Department of Agriculture and Consumer Services, Tallahassee.

- [8] Bowker, J. D., Carty, D. G., & Bowman, M. P. (2008). Inexpensive Apparatus to Rapidly Collect Water Samples from a Linear-Design, Plug-Flow Hatchery Raceway. *North American Journal of Aquaculture*, 70(1), 8–11. <https://doi.org/10.1577/a06-060.1>
- [9] Arndt, R. E., Routledge, M. D., Wagner, E. J., & Mellenthin, R. F. (2001). Influence of Raceway Substrate and Design on Fin Erosion and Hatchery Performance of Rainbow Trout. *North American Journal of Aquaculture*, 63(4), 312–320. [https://doi.org/10.1577/1548-8454\(2001\)063<0312:iorsad>2.0.co;2](https://doi.org/10.1577/1548-8454(2001)063<0312:iorsad>2.0.co;2)
- [10] Du K. Submersible plug-flow aerator comprises nano-bubble generating system, nano-bubble flow pushing aerating machine, sinking and floating device, machine case, and pushing flow tube. Chinese Patent, CN 207748926-U
- [11] Wan Y., Han H., Chen M., & Wen X. Ballast water treatment device includes oil removal sedimentation tank, oxidation pond, tower, and oil removal sedimentation tank including mixing area, air floating area, plug-flow region, and settling zone. Chinese Patent, CN 108529822-A
- 7) The aeration tubes were made of a nano-polymer material with unique physical properties. Once the air was pumped into the aeration tube, the internal pressure increased, generating micro-bubbles. If not working, the aeration orifices can shut off automatically due to water pressure, thereby preventing clogging of contaminants. Corresponding information can be found at *Page 6, Line 126-127* and *Line 132-136* in Experimental setup and method section.

Appendix C

Response to referees

Dear Editors,

Thank you so much for your E-mail letter dated May 10, 2019, regarding our manuscript **RSOS-182061.R1** entitled “*Hydrodynamics of an in-pond raceway system with an aeration plug-flow device for application in aquaculture: an experimental study*”. We are very honored that our paper can be accepted by Royal Society Open Science after minor revision. We appreciate editors and reviewers very much for their comments and suggestions. We are deeply sorry for the comments on language in the two reviews, and we have tried our best to correct the language. We have asked for polishing service according to your recommendation. Please see the editorial certificate at the end the document and the polished manuscript returned by the Charlesworth Group. We have also studied reviewer’s comments carefully and have made revision in the paper. We believe that the revisions have improved the quality of the manuscript substantially. Enclosed please find our response to the referees’ comments, and the revised manuscript.

Thank you.

Sincerely,

Xiangju Cheng.

E-mail: chengxiangju@scut.edu.cn

Response to reviewers’ comments

We thank the reviewers for their constructive comments on our manuscript. The comments are valuable and very helpful for revising and improving our paper. We have studied comments carefully and have made correction which we hope to meet with approval. The main corrections in the paper and the response to comments are as follows:

Answers to Reviewer #1

- 1) Since the plug-flow is not a general meaning in this article, it should not appear in abstract without any statement. We accepted your suggestion to avoid using the unexplained terms. We also revise the content of abstract, please see the *Lines 23-27*.
- 2) In Line 62, the original words (reaction mechanism) are our mistakes in writing. Thanks for your careful review. We have made amendments to this, please see the *Line 65*.
- 3) The flow patterns in the original aquaculture raceways are similar to the plug-flow in pipes, with uniform velocity distribution in cross sections [1]. However, there is little mixing in raceways, resulting in the water quality vary significantly from inlet to outlet. And despite improvements in the flow characteristics to increase mixing, the term “plug-flow” is still

used to describe the flow of water being pumped downstream in aquaculture raceways. We explained their differences in detail according to your comment (*Line 74-78*). We also checked the references and ensured that this term is more widely in aquaculture, especially recirculating aquaculture system, such as the publications [1]-[5] listed below.

- [1] Van Wyk, P. (1999). Principles of recirculating system design. Pages 59–98 in P. Van Wyk, M. Davis-Hodgkins, R. Laramore, K. L. Main, J. Mountain, and J. Scarpa, editors. Farming marine shrimp in recirculating freshwater systems. Florida Department of Agriculture and Consumer Services, Tallahassee.
- [2] Bowker, J. D., Carty, D. G., & Bowman, M. P. (2008). Inexpensive Apparatus to Rapidly Collect Water Samples from a Linear-Design, Plug-Flow Hatchery Raceway. *North American Journal of Aquaculture*, 70(1), 8–11. <https://doi.org/10.1577/a06-060.1>
- [3] Arndt, R. E., Routledge, M. D., Wagner, E. J., & Mellenthin, R. F. (2001). Influence of Raceway Substrate and Design on Fin Erosion and Hatchery Performance of Rainbow Trout. *North American Journal of Aquaculture*, 63(4), 312–320. [https://doi.org/10.1577/1548-8454\(2001\)063<0312:iorsad>2.0.co;2](https://doi.org/10.1577/1548-8454(2001)063<0312:iorsad>2.0.co;2)
- [4] Du K. Submersible plug-flow aerator comprises nano-bubble generating system, nano-bubble flow pushing aerating machine, sinking and floating device, machine case, and pushing flow tube. Chinese Patent, CN 207748926-U
- [5] Wan Y., Han H., Chen M., & Wen X. Ballast water treatment device includes oil removal sedimentation tank, oxidation pond, tower, and oil removal sedimentation tank including mixing area, air floating area, plug-flow region, and settling zone. Chinese Patent, CN 108529822-A

We appreciate for editors and reviewers' warm work earnestly, and hope that the correction will meet with approval.

Once again, thank you very much for your comments and suggestions!

EDITORIAL CERTIFICATE

This document certifies that the manuscript below was edited for correct English language usage, grammar, punctuation and spelling by qualified native English speaking editors at The Charlesworth Group.

Paper Title:

Hydrodynamics in an in-pond raceway system with aeration plug-flow device: experimental studies

Author:

Xiangju Cheng

Date certificate issued:

May 17, 2019

cwauthors.com